# Is Your Diffusion Model Actually Denoising?

**Daniel Pfrommer**
MIT
Cambridge, MA 02139
dpfrom@mit.edu

**Zehao Dou**
Yale University
New Haven, CT 06520
zehao.dou@yale.edu

**Christopher Scarvelis**
MIT
Cambridge, MA 02139
scarv@mit.edu

**Max Simchowitz**
CMU
Pittsburgh, PA 15213
msimchow@andrew.cmu.edu

**Ali Jadbabaie**
MIT
Cambridge, MA 02139
jadbabai@mit.edu

## Abstract

We study the inductive biases of diffusion models with a conditioning-variable, which have seen widespread application as both text-conditioned generative image models and observation-conditioned continuous control policies. We observe that when these models are queried conditionally, their generations consistently deviate from the idealized "denoising" process upon which diffusion models are formulated, inducing disagreement between popular sampling algorithms (e.g. DDPM, DDIM). We introduce *Schedule Deviation*, a rigorous measure which captures the rate of deviation from a standard denoising process, and provide a methodology to compute it. Crucially, we demonstrate that the deviation from an idealized denoising process occurs irrespective of the model capacity or amount of training data. We posit that this phenomenon occurs due to the difficulty of bridging distinct denoising flows across different parts of the conditioning space and show theoretically how such a phenomenon can arise through an inductive bias towards smoothness.

## 1 Introduction

Diffusion models (DMs) have seen widespread adoption in domains as diverse as robotic control, molecule design, and image generation from text prompts. The diffusion formalism is popular both because it enables stable training of neural generative models, via the denoising training objective, and because it offers a broad menu of mathematical and algorithmic techniques for *inference* [Albergo et al., 2023]. For example, inference can be conducted via both Stochastic Differential Equation (SDE) [Ho et al., 2020] and Ordinary Differential Equation (ODE) [Karras et al., 2022] formalisms, the latter of which can be distilled further for accelerated sampling [Song et al., 2023]. The design of these inference strategies hinges on the following fact: for a given (forward) diffusion process, there are many distinct "reverse" stochastic processes, each of which can produce the same marginal distribution over generated samples. Hence, from a sampling perspective, the various stochastic processes are in effect equivalent.

A trained diffusion model is only an imperfect neural approximation to the idealized reverse processes. Nevertheless, we might hope that this approximation is not too inaccurate. For example, even if a diffusion model does not perfectly capture the target training distribution, one might conjecture that the denoising training objective ensures that the model is at least consistent, in some appropriate sense, with the forward processes mapping its own generated samples to noise. At the very least, one would hope that as a diffusion model is trained on more data, or is conditioned on contexts that are well-represented in a training dataset, a learned diffusion model will converge towards its mathematical idealization.

39th Conference on Neural Information Processing Systems (NeurIPS 2025).

**Contributions.** In this work, we initiate the study of the inductive biases of conditional diffusional models: diffusion models whose generations depend on some context $z$. For example, the context could represent text descriptions of an image, observational inputs to a robotic control policy, or molecular properties of a protein binding target. We investigate the extent to which the path probabilities, ($p_s$, Definition 2.1) deviate from those of an idealized diffusion path ($p_s^{\text{IMCF}}$, Definition 2.4) with the same initial and terminal distribution as our learned model (note: not necessarily the ground truth data distribution). To facility this study, we introduce a novel, rigorous metric, **Schedule Deviation**, that is designed to precisely measure the extent to which a flow field induces non-denoising behavior in intermediate marginal densities.

In short, we find

> Conditional diffusion models **routinely and consistently deviate from the idealized model-consistent diffusion probability path**, $p^{\text{IMCF}}$. This effect is a direct byproduct of the inductive bias of conditional diffusion, and is strongly correlated with the discrepancy between popular diffusion samplers which, mathematically, should be equivalent in the limit of small discretization error.

In more detail, our contributions are as follows:

- We introduce **Schedule Deviation** (SD), our new metric which quantifies the extent to which a diffusion model (conditional or otherwise) deviates from the idealized diffusion probability path (Definition 3.1). We show that (SD) is closely related to the average total variation distance between path measures (Theorem 1), and that SD can be efficiently evaluated as a consequence of the transport equation (Proposition 3.1). Moreover, unlike most prior metrics used to study non-denoising behavior, SD does not require access to the true score function or training data, and can be evaluated with access to only the (potentially conditional) flow-based model.
- Using Schedule Deviation, we show that the probability path of conditional diffusion models consistently and routinely deviates from the idealized path (Figure 3, Figure 4). Our findings are consistent across toy examples, conditional image generation, and trajectory planning. Furthermore, we show that SD is often predictive of the Earth Mover Distance (EMD) between the samples generated by popular inference algorithms.
- We demonstrate the Schedule Deviation cannot be significantly ameliorated by increased model capacity or training data, and even varies significantly between different classes which are equally represented in the training data (Figure 5). Rather, we posit that SD in conditional settings arises as a natural inductive bias of conditional diffusion when interpolating between multimodal distributions.
- We provide a theoretical model (Section 4, Theorem 2 and 3) of Schedule Deviation that shows the deviation can be attributed to an inductive bias of conditional diffusion involving *smoothing* with respect to the conditioning variable. We prove that, under appropriate conditions, conditional diffusion engages in "self-guidance," combining scores from nearby points in the training data set and demonstrate that this causes deviation from the idealized denoising process.

## 1.1 Related Work

The origins of diffusion models in machine learning trace back to Sohl-Dickstein et al. [2015] and were made practical by Ho et al. [2020], Song and Ermon [2019], which show such models are capable of producing state-of-the-art image generation results. The DDIM sampling scheme [Song et al., 2020a] generalizes the reverse "denoising" process to allow for a variable level of stochasticity in the reverse sampling process, spawning a large body of work on improved sampling methodology [Bansal et al., 2024, Kong and Ping, 2021, Salimans and Ho, 2022, Permenter and Yuan, 2023].

A recent line of work has investigated closed-form diffusion models [Scarvelis et al., 2023] and their relation to phenomena observed in diffusion models, such as hallucination [Aithal et al., 2024]. Drawing on the well-appreciated inductive bias of neural networks towards low-frequency functions [Rahaman et al., 2019, Cao et al., 2019], these works highlight the importance of smoothing biases in understanding hallucination and generalization in unconditional generation, with a focus on an implicit bias towards "smoothed" versions of the "true" denoiser of the training data.

A concurrent line of works [Vastola, 2025, Bertrand et al., 2025] have also investigated the non-denoising properties of diffusion models, with a particular emphasis on the role of target stochasticity in potentially inducing non-denoising behavior. However, these works focus on deviation from the

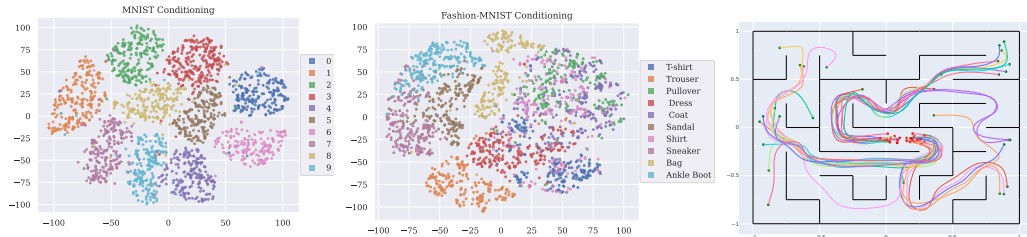

Figure 1: We principally consider three datasets: conditional MNIST [LeCun et al., 1998] (left), conditional Fashion-MNIST [Xiao et al., 2017] (middle), and endpoint-conditional maze path generation (right). For MNIST and Fashion-MNIST we condition on the t-SNE embedding of the images (pictured above) as opposed to the classes as a proxy for text-embedding-conditioned image generation.

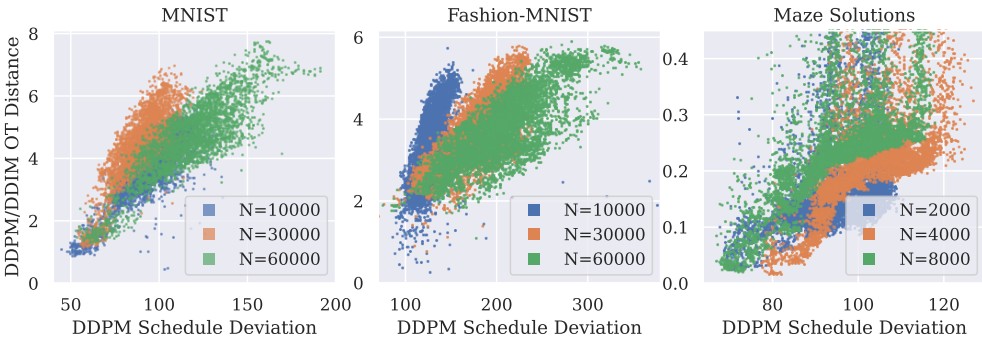

Figure 2: For conditioning values $z \sim \text{Unif}(\mathcal{Z})$, we plot the Total Schedule Deviation (for $p_0$ sampled using DDPM) and optimal transport distance between DDPM/DDIM samples (as measured by 1-Wasserstein/Earth-Mover-Distance), demonstrating that our prposed metric, Schedule Deviation, is indeed predictive of divergence between different samplers. In Appendix C we demonstrate these trends hold across different choices of samplers and show additional experiments on attribute-conditional Celeb-A, where the conditioning space is more uniform.

true denoiser specified by the training data, whereas we consider denoising self-consistency of the model itself. In this regard our work is similar to Daras et al. [2024], which introduces an additional loss term to induce better denoiser self-consistency. Unlike Vastola [2025] and Bertrand et al. [2025], which reach differing conclusions on whether deviation from an ideal denoiser benefits sample quality, we take no position on whether non-denoising behavior is beneficial.

Our work is orthogonal yet complementary to these efforts, elucidating the effect of inductive biases for *conditional* diffusion models [Ho et al., 2022, Song et al., 2020b]. Unlike the unconditional setting, where generalization necessitates underfitting the training data, we show that an implicit bias towards smoothness with respect to the conditioning variable can bias the model to out-of-distribution conditioning variables while simultaneously overfitting the training data. Of particular interest to this work is the methodology of classifier-free guidance [Ho and Salimans, 2022], which performs conditional sampling via a combination of both conditional and unconditional diffusion models. Although we consider purely conditional models, we show that under the appropriate implicit biases, a form of classifier-free guidance, which we term "self-guidance," can naturally arise. The exact effect of guidance on diffused samples remains the subject of ongoing inquiry, but recent work has shown that special forms of guidance can be (approximately) understood as either sampling from the manifold of equiprobable density [Skreta et al., 2024] or, alternatively, as a combination of Langevin dynamics on the weighted product distribution [Bradley and Nakkiran, 2024, Shen et al., 2024]. We do not further investigate the mechanism by which guidance yields high quality intermediate samples, but rather demonstrate on simple datasets that the form of "self-guidance" has predictive capabilities.

Prior work has shown that classifier-free guidance causes the resulting diffusion to no longer constitute a denoising process [Bradley and Nakkiran, 2024]. We note that our notion of consistency is unrelated to that of Consistency Models [Song et al., 2023], which may in fact be schedule-inconsistent under our framework. Daras et al. [2024] explore a similar notion of consistency with a denoiser and attempt to enforce this condition by a data-augmention-style loss. In contrast, our metric is tailored to specifically measure how much the evolved density diverges from a reference denoising process, as opposed to how much it deviates from being a diffusion model.

## 2 Preliminaries

We apply the framework for flow-based generated models in Lipman et al. [2022] and Albergo et al. [2023], adapted to our conditional setting. For mathematical elegance, we adopt a continuous-time formalism; see e.g. Chan et al. [2024] for the discrete-time formalism. We consider pairs $(x, z)$, where $z \in \mathcal{Z} \subset \mathbb{R}^{d_z}$ is a conditioning value and $x \in \mathcal{X} \subset \mathbb{R}^{d_z}$ is a datapoint. For a given conditioning value $z$, we seek to generate samples from a continuous conditional distribution $p^\star(x|z)$. For instance, $x$ may be an image or action while $z$ can be a text prompt or observation. We use $Z, X$ to denote random variables over $\mathcal{Z}$ and $\mathcal{X}$, respectively, and $z \in \mathcal{Z}, x \in \mathcal{X}$ for particular values.

**Flow-based generative models** parameterize a time-varying family of conditional densities $p_s(x|z), s \in [0, 1]$, where $p_1(x|z)$ can be easily sampled, and $p_0(x|z)$ a conditional density aimed to approximate $p^\star$. These densities can be specified *conditional normalizing flows* [Lipman et al., 2022], which describe the per-time-step marginals $p_s$ via the evolution of particles moving according to a specified velocity field $v_s(x, z)$.

**Definition 2.1** (Probability Paths and Conditional Flows). Let $s \in [0, 1]$ be a time index. Fix a flow field $v : [0, 1] \times \mathcal{X} \times \mathcal{Z} \to \mathcal{X}$ and a family of conditional densities $p_s(x|z)$ indexed by time $s \in [0, 1]$, which we refer to as a *probability path*. We say $(v, p)$ is a **conditional normalizing flow** if particles evolved under $v$ match the marginal densities $p_s$ of the probability path, i.e..

$$p_s(\cdot|z) = \mathrm{Law}(X_s \mid Z = z), \ \ \forall s \in [0, 1], z \in \mathcal{Z}, \tag{2.1}$$

$$\text{where } X_1|Z = z \sim p_1(\cdot|z), \frac{\mathrm{d}}{\mathrm{d}s}X_s = v_s(X_s, z).$$

The **stochastic interpolants** framework [Albergo et al., 2023] provides an alternative description for flow-based models, with a particularly succint description of diffusion models (e.g. Ho et al. [2020]).

**Definition 2.2.** A **diffusion schedule** $(\sigma, \alpha)$ consists of a noise schedule $\sigma : [0, 1] \to \mathbb{R}_{\geq 0}$ and signal schedule $\alpha : [0, 1] \to \mathbb{R}_{\geq 0}$ such that $\sigma(1) = \alpha(0) = 1$, and $\alpha(1) = \sigma(0) = 0$.

**Definition 2.3** (Diffusion Probability Path). Given a diffusion schedule $(\sigma, \alpha)$, target and initial densities $p_0$ and $p_1$, the **diffusion probability path** is given by,

$$p_s(\cdot|z) = \mathrm{Law}(X_s|Z_s = z)$$

where $X_s := \alpha(s)X_0 + \sigma(s)X_1$, denotes the **stochastic interpolant** [Albergo et al., 2023], where $X_1 \mid Z = z \sim p_1 = \mathcal{N}(0, I)$ and $X_0|Z = z \sim p_0(\cdot|z)$ with $X_1 \perp X_0|Z$.

**Model-Consistent Diffusion Flows** Under appropriate regularity conditions, all diffusion probability paths can be realized by an infinite family of conditional normalization flows $(v, p)$. We focus on one such path, the *model-consistent diffusion flow*, which corresponds to the solution of the DDPM objective [Ho et al., 2020]. We use $\hat{v}$ and $\hat{p}$ throughout the rest of this paper to emphasize conditional normalizing flows which may be associated with a learned model, rather than the true data-generating distribution, i.e. where $\hat{p}_0 \neq p^\star$.

**Definition 2.4** (Ideal Model-Consistent Flow). Fix a (potentially learned) flow and associated family of densities $(\hat{v}, \hat{p})$ as defined in Definition 2.1. For a given diffusion schedule $(\sigma, \alpha)$, let $p^{\mathrm{IMCF}} = (p_s^{\mathrm{IMCF}}(x|z))_{s \in [0,1]}$ be the diffusion probability path associated with $\hat{p}_0$. The ideal denoising diffusion flow (IMCF) of $\hat{p}$, written $v^{\mathrm{IMCF}} = \mathrm{IMCF}(\hat{p}_0)$, is the unconstrained global minimizer of

$$L[v] := \mathbb{E}[\|v_s(X_s, Z) - (\dot{\alpha}(s)X_0 + \dot{\sigma}(s)X_1)\|^2]. \tag{2.2}$$

Above, the expectation $\mathbb{E}$ is taken w.r.t. the diffusion path Definition 2.3 with $s \sim \mathrm{Unif}([0, 1])$. The IMCF is unique, and furthermore the optimization in (2.2) can be decoupled across conditioning

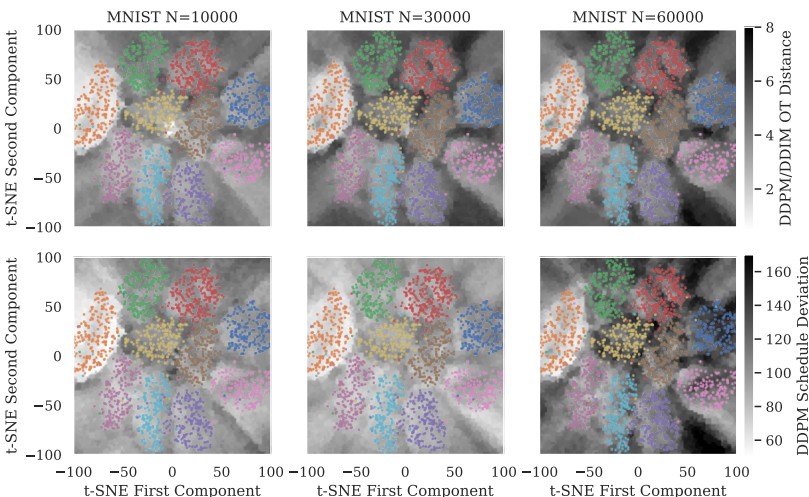

Figure 3: For t-SNE-conditional MNIST generation, we evaluate the Schedule Deviation and empirical 1-Wassertstein Distance between DDPM/DDIM samples, ablated over the training dataset size $N \in \{10000, 30000, 60000\}$. We note strong structural similarity between the two metrics that appears related to the contours of the conditioning distribution and the conditional data distributions.

variables $z$. Thus, we will often write $v^{\text{IMCF}}(\cdot, z) = \text{IMCF}(\hat{p}_0(\cdot \mid z))$ for a fixed value of $z$. The IMCF corrresponds to the unique velocity-minimizing flow consistent with the diffusion probability path, which can be characterized explicitly as follows (see Appendix A.1.1).

**Remark 2.1** (Ideal Model-Consistent Flow vs Ground Truth Flow). We use $\hat{p}$ instead of $p$ in Definition 2.4 from here on to emphasize that $\hat{p}_0 \neq p^\star$, meaning that $v^{\text{IMCF}}$ is the ideal flow associated with the distribution of *learned model* $\hat{v}$ under a given sampling algorithm; not necessarily the "true" flow $v^\star$. Hence, schedule deviation is potentially orthogonal as a metric to whether $p_0$ matches $p^\star$. However, we note that, by the construction of $v^{\text{IMCF}}$, $\hat{p}_0 = p_0^{\text{IMCF}}$ and $\hat{p}_1 = p_1^{\text{IMCF}}$.

**Proposition 2.1.** *Adopt the setup of Definition 2.4. Then $v = v^{\text{IMCF}} = IMCF(p_0)$ is given explicitly by any of the following identities*

$$v_s^{\text{IMCF}}(x, z) = \mathbb{E}[\dot{\alpha}(s)X_0 + \dot{\sigma}(s)X_1 | X_s = x, Z = z] \tag{2.3}$$

$$= \gamma_1(s)\nabla_x \log p_s(x|z) + \gamma_2(s)x. \tag{2.4}$$

$$= c_1(s)\mathbb{E}[X_0 | X_s = x, Z = z] + c_2(s)x \tag{2.5}$$

*where $c_1(s) := \dot{\alpha}(s) - \frac{\dot{\sigma}(s)}{\sigma(s)}\alpha(s)$, $c_2(s) := \frac{\dot{\sigma}(s)}{\sigma(s)}$, $\gamma_1(s) := \frac{\dot{\alpha}(s)}{\alpha(s)}\sigma(s)^2 - \dot{\sigma}(s)\sigma(s)$, $\gamma_2(s) := \frac{\dot{\alpha}(s)}{\alpha(s)}$.*

Eq. (2.4) represents the MCF in terms of the *score functions* $\nabla_x \log p_s(x \mid z)$, whereas Eq. (2.5) expresses the flow in the classical "denoising" form [Ho et al., 2020] via the conditional expectation of the "clean" datapoint $X_0$ given the "noised" $X_s = \alpha(s)X_0 + \sigma(s)X_1$. Hence, the IMCF $v^{\text{IMCF}}$ is just the (continuous-time) denoising objective from DDPM [Ho et al., 2020].

## 3   Conditional Diffusion is Not Denoising.

This section introduces our first main finding: **the probability paths in diffusion consistently deviate from the idealized model-consistent probability path** $p^{\text{IMCF}}$, often in regions with *high data density*, and in a manner that *does not abate with increased number of training examples*.

### 3.1   Measuring Schedule Deviation

To quantify this effect, we start by introducing **Schedule Deviation**, a novel, natural metric which evaluates the extent to the flow field $v$ induces instantaneous deviations from the idealized probability path $p^{\text{IMCF}}$ associated with $v^{\text{IMCF}} = \text{IMCF}(\hat{p}_0)$. Crucially, Schedule Deviation analyzes the behavior of the learned model $\hat{v}$ on $p^{\text{IMCF}}$, that is, the forward process associated with the distribution $\hat{p}_0$ generated by the model and not the reverse process $\hat{p}_t$ itself (as in Daras et al. [2024]).

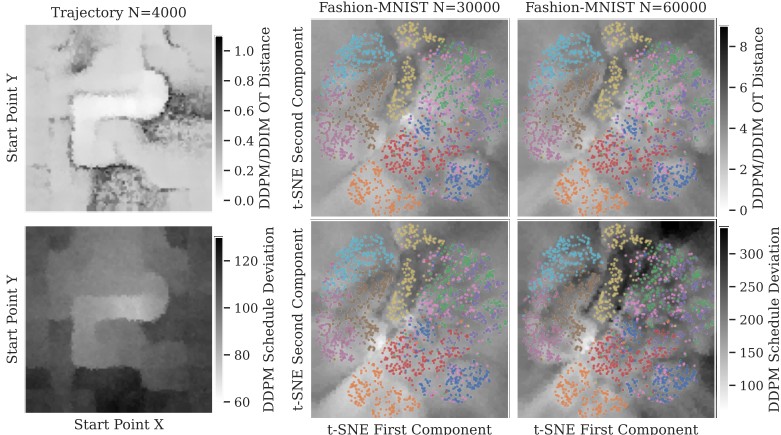

Figure 4: Analogous to Figure 3, we show that Schedule Deviation is predictive of divergence between the DDPM/DDIM samplers for the trajectory (left) and Fashion-MNIST datasets (right). Note that the structure of the maze (shown in Figure 1) can clearly be observed in the Schedule Deviation. We defer full ablations over the training data for both to Appendix C.

**Definition 3.1** (Schedule Deviation). Fix a diffusion schedule $(\alpha, \sigma)$ and conditional flow model $(v, p)$ where $p_1(x|z) = \mathcal{N}(0, I)$. Let $p_s^{\text{IMCF}}(x|z)$ be the diffusion probability path (Def. 2.3) for $p_0(x|z)$ and consider the tangent probability path $p_{s|t}^v = \text{Law}(X_{s|t}^v)$ which begins at time $t$ with $X_{t|t}^v \sim p_t^{\text{IMCF}}(x)$ and evolves as $\frac{\mathrm{d}}{\mathrm{d}s} X_{s|t}^v = v_s(X_s)$. We define the **Schedule Deviation** at $z \in \mathcal{Z}, s \in [0, 1]$ as

$$\text{SD}(v; z, s) := \int_{\mathcal{X}} \left| \left[ \frac{\partial p_{s|t}^v}{\partial s} \right]_{t=s} - \frac{\partial p_s^{\text{IMCF}}}{\partial s} \right| dx.$$

We additionally define the Total Schedule Deviation of $v$ at $z \in \mathcal{Z}$ by $\text{SD}_{\text{total}}(v; z) := \int_0^1 \text{SD}(v; z)\mathrm{d}s$. Where appropriate, we simply refer to $\text{SD}_{\text{total}}$ as the Schedule Deviation at a given $z \in \mathcal{Z}$.

We refer to the random variable $X_{s|t}^v$ as a *tangent* process [Falconer, 2003] because it initially coincides with $p_s^{\text{IMCF}}$ at $s = t$, but the evolves differently according to the flow field dictated by $v$. Schedule Deviation (SD) therefore measures the rate at which a learned flow instantaneously departs from the IMCF associated with *its own* data generating distribution $p_0$.

In addition to its natural relationship to the instantaneous deviation from the IMCF, Schedule Deviation can also be tractably estimated (proved in Appendix A.3.2).

**Proposition 3.1.** *It holds that* $\text{SD}(v; z, s) = \mathbb{E}_{p_s^{\text{IMCF}}}[|\nabla \cdot (v_s - v_s^{\text{IMCF}}) + (v_s - v_s^{\text{IMCF}}) \cdot \nabla \log p_s^{\text{IMCF}}|]$.

For $z, s$ fixed, we can directly sample from $p_s^{\text{IMCF}}$ and estimate the score $\nabla \log p_s^{\text{IMCF}}(x|z)$, and consequently, the idealized flow $v_s^{\text{IMCF}}(x, z)$, by constructing an empirical distribution $\{x_0^i\}_{i=1}^n$ sampled from $p_0(x|z)$, convolving each $x_0^i$ with Gaussian noise as dictated by the Diffusion Schedule, and estimating the corresponding score in closed form. See Algorithm 1 for high-level pseudodocode for computing Schedule Deviation given a particular sampling algorithm using $n$ independent estimates using $n \cdot (N + 1)$ total samples and Appendix C for implementation details. Importantly, because we evaluate $\nabla \log p_s^{\text{IMCF}}(x|z)$ for $z, s$ fixed, our estimator is not subject to the inductive biases of function approximation that induce schedule deviation in the learned neutral network.

---

**Algorithm 1** Schedule Deviation

**input:** $z \in \mathcal{Z}, s \in [0, 1]$
$R \leftarrow \emptyset$
**for** $i \in \{1, \dots, n\}$ **do**
  $x_0 \leftarrow \text{sample}(v, z)$
  $x_s \leftarrow \alpha(s)x_0^{(i)} + \sigma(s)\epsilon$ where $\epsilon \sim \mathcal{N}(0, I)$
  $S \leftarrow \emptyset$
  **for** $j \in \{1, \dots, N\}$ **do**
    $S \leftarrow S \cup \{\text{sample}(v, z)\})$
  **end for**
  compute $\nabla \log p_s^{\text{IMCF}}, v_s^{\text{IMCF}}, \nabla \cdot v_s^{\text{IMCF}}$
    using Eq. (2.3), $p_0(x_0|z) \approx \frac{1}{N}\sum_{x \in S} \delta_x$
  $r_1 \leftarrow \nabla \cdot [v_s - v_s^{\text{IMCF}}](x_s, z)$
  $r_2 \leftarrow [v_s - v_s^{\text{IMCF}}](x_s, z) \cdot \nabla \log p_s^{\text{IMCF}}(x_s|z)$
  $R \leftarrow R \cup \{r_1 + r_2\}$
**end for**
**return** $\text{mean}(R)$

---

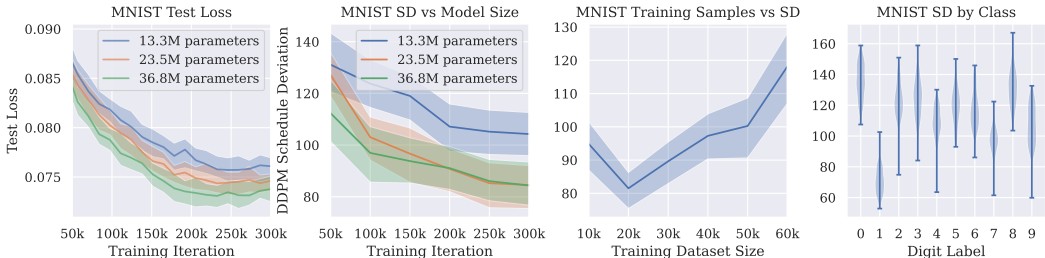

Figure 5: We visualize the test loss (left) and total schedule deviation (center left) for three different model capacities over the course of a training run. For the 13.3M parameter model, we show the effect of training dataset size on schedule deviation (center right), and, for the full dataset, the distribution of total schedule deviation across different classes (right). The median, 30th, and 70th percentile values are shown across the left three plots for sampled training batches and conditioning values.

The definition of Schedule Deviation closely resembles the formulation of the classical Performance Difference Lemma in Reinforcement Learning [Kakade and Langford, 2002]. Indeed, under appropriate smoothness assumptions, the schedule deviation both upper and lower bounds the total variation difference in the probability paths $p^{\text{IMCF}}$ and $p$. We defer the proof to Appendix A.3.1.

**Theorem 1.** *Consider the setting of Definition 3.1, for a conditional flow $(v, p)$. For any probability measure $\mu$ over $[0, 1]$, the total variation distance $\text{TV}(q, p) := \frac{1}{2} \int |p - q| \mathrm{d}x$ between $p_s$ and $p_s^{\text{IMCF}}$ over $\mu$ is upper bounded by integrated schedule deviation:*

$$\int \text{TV}(p_s(\cdot|z), p_s^{\text{IMCF}}(\cdot|z)) d\mu(s) \leq \int_0^1 \text{SD}(v; z, t) \mathrm{d}t = \text{SD}_{\text{total}}(v, z).$$

*Moreover, if $|\partial_s^2 p_s^{\text{IMCF}}|$, $|\partial_{st}^2 p_{s|t}^v|$, $|\partial_s^2 p_s| \leq M < \infty$, there exists constants $\epsilon_0 \in [0, 1/2], c \geq 0$ depending on $M$ such that, for all $0 < \epsilon \leq \epsilon_0$, $s \in [0, 1]$*

$$\sup_{t \in [s-\epsilon, s+\epsilon] \cap [0,1]} \text{TV}(p_t(\cdot|z), p_t^{\text{IMCF}}(\cdot|z)) \geq c\epsilon \text{SD}(v; z, s). \tag{3.1}$$

**Remark 3.1** (Schedule Deviation v.s. Generation Fidelity). Note that $p_0 \neq p^\star$, so $v^{\text{IMCF}}$ is the IMCF associated with the distribution of *learned model* under a given sampler; not necessarily the IMCF associated with the true data distribution $p^\star$. Hence, schedule deviation is potentially orthogonal as a metric to whether $p_0$ matches $p^\star$. Moreover, we note that $p_0 = p_0^{\text{IMCF}}$ and $p_1 = p_1^{\text{IMCF}}$, so that schedule deviation principally captures deviations from the reference process in the "middle" of the denoising.

**Remark 3.2.** It should be noted that Schedule Deviation is distinct from consistency distillation [Song et al., 2023]. Consistency distillation enforces a related condition: that a few-step model is consistent with the integrated flow map of the ODE induced by the flow-field $v$, whereas Schedule Deviation measures the deviation of the flow map from the denoising probability path.

## 3.2 Schedule Deviation is Widely Prevalent

We evaluate the schedule deviation of trained neural networks in two distinct settings and 3 datasets, as described below. We use a U-Net architecture similar to Dhariwal and Nichol [2021] for all experiments. For full experiment details, see Appendix C.

**Setting 1 (Conditional Image Generation).** We evaluate the schedule-deviation of conditional image diffusion models. For ease of visualization and in order to keep the associated dimensionality of $x$ low (as Definition 3.1 requires computing the divergence w.r.t. $x$), we consider the **MNIST** [LeCun et al., 1998] and **Fashion-MNIST** [Xiao et al., 2017] datasets, each conditioned on a 2-dimensional "latent" obtained via a t-SNE [Van der Maaten and Hinton, 2008] embedding of the data. In Appendix C.1, we additionally consider a much larger model trained on the **Celeb-A** dataset, using a t-SNE of the discrete attribute space for the conditioning variable. We note that the Celeb-A experiments, which use a simplified Schedule Deviation without the divergence component, are much more noisy and less conclusive than the corresponding MNIST or Fashion-MNIST experiments.

**Setting 2 (Conditional Maze Paths).** Second, we construct a simplified path-planning problem consisting of generating trajectories in a fixed maze. For a given randomly chosen starting point, we consider all paths $\{r_i\}_{i=0}^{K}$ to the center of the maze and sample a path $r_i$ with probability $p(r_i) \propto e^{-(d(r_i)-d(r^\star))}$, where $d(r_i)$ is the length of the $i$th path and $d(r^\star)$ is the length shortest path. This artificially introduces multimodality around points where multiple solutions are approximately equally optimal. For each sampled path, we use 64 points along smooth Bezier curve fit to the path such that $\mathcal{X} \subset \mathbb{R}^{64 \times 2}, \mathcal{Z} \subset \mathbb{R}^2$.

**Finding 1: Prevalence of Schedule Deviation.** Our experiments broadly demonstrate that Schedule Deviation is prevalent across all datasets. We visualize the total Schedule Deviation over $z \in \mathcal{Z}$ in Figure 3 and Figure 4 for each of our datasets with varying subsets of the training data.

**Finding 2: Schedule Deviation Persists with Model Size and Data Amount.** In Figure 5, we explore the schedule deviation for the MNIST dataset in-depth for both model and data ablations. Interestingly, we find that while larger models tend to exhibit slightly lower Schedule Deviation-—the improvements appear to diminish as the model size increases and more training data can potentially (and somewhat counter-intuitively) *increase* the Schedule Deviation. Furthermore Figure 5 (right), shows that the Schedule deviation can vary dramatically between different classes, suggesting that the Schedule Deviation is both a function of the density and structure of underlying dataset.

**Key Takeaways:** Our experiments indicate several key properties on the Schedule Deviation of conditional diffusion models: (1) even high-capacity models can exhibit significant Schedule Deviation, (2) Schedule Deviation appears to be intrinsically related to the underlying structure of the dataset, rather than the amount of data, and, perhaps most importantly, and (3) as we will show in the following section, Schedule Deviation is strongly predictive of divergence between different samplers.

### 3.3 Schedule Deviation Predicts Disagreement Between Samplers

Many popular sampling algorithms, such as DDPM [Ho et al., 2020] and DDIM [Song et al., 2020a] leverage an SDE formalism to sample from the target distribution $p^\star$ (see Appendix B for details). These sampling algorithms implicitly make use of the equivalence in Proposition 2.1 between the learned flow $v$ and $\nabla \log p_s(x|z)$ to traverse the same denoising probability path with differing levels of noise in the reverse process. Thus, when $v = \text{IMCF}(p_0)$, i.e. there is no schedule deviation, both are guaranteed to generate samples $X_s$ whose marginals coincide with the conditional flow in Definition 2.1 (provided the number of steps is sufficiently large that discretization error is negligible).

Empirically, significant task-specific differences in performance between DDIM and DDPM have been observed Chi et al. [2023], Song et al. [2020a], Karras et al. [2022]. In Figure 2, we show that Schedule Deviation is strongly correlated with the difference between these samplers, as measured by the empirical 1-Wasserstein (i.e. Earth-Movers-Distance). The heatmaps Figure 3, Figure 4 further demonstrate the structural similarity of Schedule Deviation and DDPM/DDIM divergence across the conditioning space. Recall Theorem 1, which confirms that SD is a proxy for the TV distance between the traversed path the ideal denoising path. Taken together with the strong correlation between SD and OT Distance (Figures 3 and 4), we conclude

> **DDPM and DDIM deviate specifically for conditioning values where the trained diffusion model deviates from the idealization of denoising its generations**.

We believe that this finding both (1) sheds light on the underlying cause for sampler divergence and (2) demonstrates the utility of our proposed metric as an investigatory tool. In Appendix C, we show our metric is predictive for other sampling strategies, such as the Gradient-Estimation (GE) sampling algorithm [Permenter and Yuan, 2023].

## 4 Explaining Schedule Deviation via Smoothness and Self-Guidance

Generalization in unconditional diffusion is broadly understood as a phenomena that arises from capacity-related underfitting of the empirical score function [Yoon et al., 2023, Scarvelis et al., 2023], thereby preventing memorization of the training data. Prior work in this area has examined the effect of an implicit bias towards *smoothness* and its relation to generalization [Scarvelis et al., 2023, Pidstrigach, 2022, Aithal et al., 2024]. These works, however, do not fundamentally challenge the assumption that the learned flows denoise and instead show how "better" denoisers arises by manifold

learning [Pidstrigach, 2022] or interpolation over convex hulls of the data [Scarvelis et al., 2023]. In fact, as we elucidate in Appendix B, the "natural" nonparametric extension of the closed-form in [Scarvelis et al., 2023] constitutes an ideal flow. In this section, we provide intuition for how *non-denoising paths* can arise from smoothness with respect to the conditioning variable through a phenomena we term **self-guidance**.

**Self-Guidance and Schedule Deviation under Discrete Support.** We begin by considering the special case where the data generating distribution $p^\star(z)$ is supported by a discrete set $S_z$. We can observe in Figure 6 that for the discrete conditioning distribution shown, the schedule deviation is almost uniformly 0 for $z \in S_z$. Thus, we motivate the following assumptions: (1) the model has sufficient capacity to exactly fit the training data everywhere where there is conditioning support, (2) the model is second-order smooth with respect to $z$, and (3) of all flows which fit on the support of the training data, the model generalizes via the "smoothest" flow, as measured by $\|\nabla_z^2 v_s(x,z)\|_{\mathcal{L}_2}$. Under these assumptions, we develop the following result for $\mathcal{Z} \subset \mathbb{R}$, (with proof in Appendix A.4.1):

**Theorem 2** (Discrete-support Smooth Interpolant). *Let $Z$ be supported on a finite set $S_z = \{z^{(i)}\}_{i=1}^N \subset \mathbb{R}$ for distinct $z^{(1)} < \ldots < z^{(N)}$, ordered without loss of generality. For each $z^{(i)} \in S_z$, let $v_s^\star(x, z^{(i)}) := \mathbb{E}[\dot{\alpha}(s)X_0 + \dot{\sigma}(s)X_1 | Z = z^{(i)}]$. Then, there are piecewise cubic polynomials $p^{(i)}(z)$, with pieces defined by the intervals $[z^{(j)}, z^{(j+1)}]$ such that*

$$v_s(x,z) = \operatorname*{arg\,inf}_{v \in \arg\inf_v L[v]} \int \|\nabla_z^2 v\|_F^2 \mathrm{d}x = \sum_{z^{(i)} \in S_z} p^{(i)}(z) \cdot v_s^\star(x, z^{(i)}).$$

*In the case where $|S_z| = 2$, $p^{(i)}(z)$ are linear functions.*

**Remark 4.1.** Because diffusion models exhibit both smoothness biases in both $x$ (e.g. Aithal et al. [2024]) and $z$ (this work), the most comprehensive proxy would consider a smoothness penalty on the joint Hessian $\nabla_{x,z}^2 v(x,z)$. This makes a closed form solution considerably more involved, and thus we focus solely on the $\nabla_z$ effect to isolate functional dependence on $z$.

The optimal low-$z$-curvature flow characterized in Theorem 2 extrapolates to out-of-distribution variables $z$ to by linearly combining flows associated with in-distribution variables $z_i \in S_z$, with weights depending *only* on the conditioning variable $z$. We refer to the phenomenon of extrapolating via combinations of flows from other parts of the conditioning space as **self-guidance**, as these linear combinations of flows mirrors the practice of classifier free guidance [Ho and Salimans, 2022], which composes conditional and unconditional flows $v(x|z), v(x)$ via a linear combination.

**Schedule Deviation Emerges from Smoothing.** Linear combinations of flows in general cannot be written as denoising flows (e.g. in classifier guidance, [Bradley and Nakkiran, 2024]). In particular, for unconditional diffusion probability paths $p_s(x|z = z^{(i)})$, linearly combining $v_s^\star(x, z^{(i)}) = \gamma_1(s)\nabla \log p_s(x|z = z^{(i)}) + \gamma_2(s)x$ (recall Proposition 2.1) does not yield a diffusion probability path, i.e. for weights $c_1, c_2$,

$$c_1 \nabla \log p_s^{(i)} + c_2 \nabla \log p_s^{(j)} = \nabla \log \left( (p_s^{(i)})^{c_1}(\hat{p}_s^{(j)})^{c_2} \right) \neq \nabla \log[(p_0^{(i)})^{c_1}(p_0^{(j)})^{c_2}]_s$$

where $[\tilde{p}]_s$ denotes the distribution of $X_s := \alpha(s)X_0 + \sigma(s)X_1$ in Definition 2.3 under $\mathrm{Law}[X_0] = \tilde{p}$. Thus, Theorem 2 suggests that a simple inductive bias towards smoothness can naturally lead to schedule deviation in $v$ when the interpolated $v_s^{\mathrm{IMCF}}(x, z^{(i)})$. We show specifically how the schedule deviation arises for particular choices of diffusion probability paths, $p_s^{(i)}, p_s^{(j)}$ in Appendix B.

**Self-Guidance with Uniform Conditioning**. We additionally show that self guidance can occur even in the presence of continuous densities, where the minimize $v^\star$ of Eq. (2.2) is uniquely specified by considering a $\lambda$-weighted penalty term with respect to the Frobenius norm of the appropriate Hessian:

$$L_\lambda[v] = L[v] + \lambda \cdot \mathbb{E}_Z \mathbb{E}_{X_s|Z}[\|\nabla_z^2 v_s(X_s, Z)\|_{\mathrm{F}}^2], \tag{4.1}$$

where $L[v]$ is the original loss in Eq. (2.2) above and the expectation $X_s \mid Z$ is taken with Definition 2.3, whose density we recall is $p_s^{\mathrm{IMCF}}(x, z)$.

**Theorem 3.** *Fix some diffusion schedule $(\alpha, \sigma)$ and let $v_s^\star(x, z)$ be the IMCF flow associated with $p^\star(x|z)$, i.e. the minimizer of Eq. (2.2). Assume that $p^\star(z)$ is a uniform density over some set $S$, i.e. $p^\star(z) = c \cdot 1_S$ for some $c > 0$ and where $1_S$ the characteristic function of $S$. Then the minimizer to $L_\lambda[v]$ for any $z \in S$ is given by,*

$$v_s(x,z) = \int_{\xi, z' \in S} \frac{e^{2\pi i \xi(z - z')}}{1 + \lambda \|\xi\|^4} v^\star(x, z') \mathrm{d}z' \mathrm{d}\xi. \tag{4.2}$$

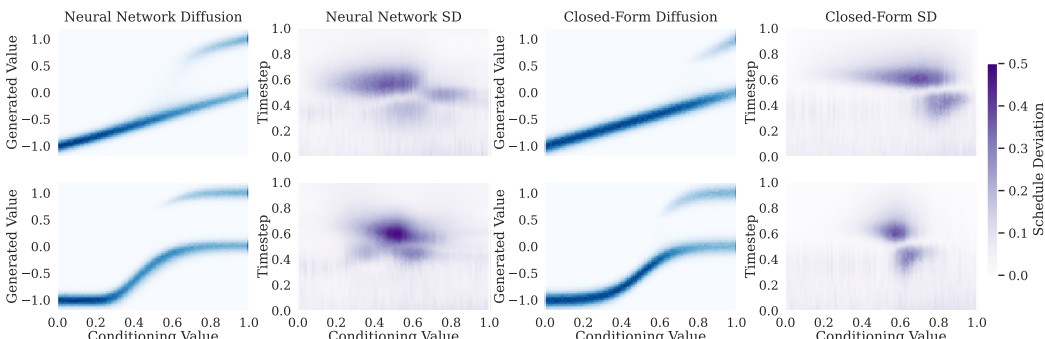

Figure 6: We train an MLP and construct a closed-form denoiser for both the discrete-support dataset (top row) and continuous-support dataset (bottom row) described below. For both the NN and closed-form interpolator we show generated values across $z \in [0, 1]$ as well as the Schedule Deviation over time. In particular, we note that for each dataset the NN and its closed-form analogue exhibit similar inductive biases as well as Schedule Deviation away from the training data.

In the case of the uniform densities, Theorem 3 reveals that self-guidance occurs via a local convolution with Fourier-weights $\int \frac{1}{1+\lambda\|\xi\|^4} e^{i\xi \cdot (z-z')} d\xi$. The frequencies $\xi$ are attenuated polynomially as $1/\|\xi\|^4$. As $\lambda \to 0$, the attenuation is removed, and the integral $\int e^{i\xi \cdot (z-z')} d\xi$ behaves like a Dirac $\delta$ around $z$ [Duistermaat et al., 2010, Chapter 14]. This passes the "sanity check" that, as the smoothness penalty vanishes, averaging becomes ever more local.

**Toy Datasets.** Motivated by Theorem 2 and Theorem 3, we consider two synthetic datasets with scalar $x \in \mathcal{X} \subset \mathbb{R}$ and condition $z \in \mathcal{Z} \subset [0, 1]$ consisting of mixtures with components centered at $\mu \in \{(0, -1), (1, 0), (1, 1)\}$. The first dataset (with "discrete support") has Gaussian noise with scale $\sigma = 0.1$ applied only to the $x$ component, while the second (with "continuous support") has IID noise of magnitude $\sigma$ applied to both the $x, z$ components.

We visualize these datasets and samples from a learned denoiser, as well as a closed-form interpolants, in Figure 6. For the discrete-conditioning setting, our closed form interpolant considers a simple linear guidance-style interpolation of the flows at $z = 0$ and $z = 1$. In the continuous-conditioning setting, inspired by the Fourier-convolution weighting in Theorem 3, we construct an interpolation with a nonlinear guidance function. See Appendix C for additional details. These experiments validate that self-guidance can be a fundamental primitive for extrapolation in conditional settings and predict the learned behavior of neural networks.

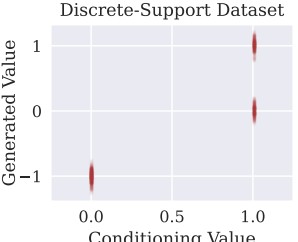

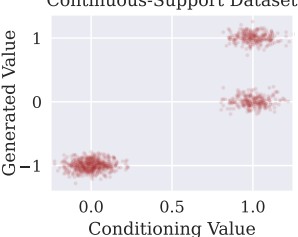

Figure 7: Training data samples from each of the toy datasets used in Figure 6.

## 5 Discussion

We introduce *Schedule Deviation*, a novel, principled metric for measuring divergence of diffusion models from their idealized denoising paths. This metric is strongly predictive of deviation between different samplers and is difficult to ameliorate via increased model capacity and data quantity. Taken together, our findings reveal that **the central mathematical abstraction upon which equivalent inference algorithms are derived may not be representative of actual diffusion models trained in practice**, and the breakdown thereof cause seemingly equivalent methods to differ. This finding has major implications for the development of future sampling and distillation methods, and serves as a broader word of caution for the use of mathematical principles, in isolation, as a sole basis for algorithm design. However, our study has a number of **limitations** (see Appendix D: in short, our metric requires computing the divergence of the flow over generated samples, an inherently expensive operation).

## Acknowledgments

DP and AJ acknowledge support from the Office of Naval Research under ONR grant N00014-23-1-2299 and the DARPA AIQ program. DP additionally acknowledges support from a MathWorks Research Fellowship. MS was supported by a Google Robotics Research Grant.

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

# A  Deferred Proofs

In this section, we provide the omitted proofs of all propositions and theorems in our main text.

## A.1  Appendix for Preliminaries (Section 2)

In addition to estabilishing the characterization in Proposition 2.1, we also show an additional fact that aids in our interpretation: the IMCF flow minimizes Euclidean velocity.

**Proposition A.1.** $v = IMCF(p)$ *is the unique flow minimizing the objective*

$$\tilde{L}[v] := \mathbb{E}_{Z, s \sim \mathrm{Unif}[0,1]} \mathbb{E}_{X_s | Z}[\|v_s'(X_s, Z)\|^2],$$

*subject to the constraint that $(v, p)$ is a conditional normalizing flow.*

The proof of this statement and Proposition 2.1 rely on the following standard transport equation, for a flow.

### A.1.1  Proof of Proposition 2.1, Proposition A.1

**Proposition 2.1.** *Adopt the setup of Definition 2.4. Then $v = v^{\mathrm{IMCF}} = IMCF(p_0)$ is given explicitly by any of the following identities*

$$v_s^{\mathrm{IMCF}}(x, z) = \mathbb{E}[\dot{\alpha}(s)X_0 + \dot{\sigma}(s)X_1 | X_s = x, Z = z] \tag{2.3}$$

$$= \gamma_1(s)\nabla_x \log p_s(x|z) + \gamma_2(s)x. \tag{2.4}$$

$$= c_1(s)\mathbb{E}[X_0 | X_s = x, Z = z] + c_2(s)x \tag{2.5}$$

*where $c_1(s) := \dot{\alpha}(s) - \frac{\dot{\sigma}(s)}{\sigma(s)}\alpha(s)$, $c_2(s) := \frac{\dot{\sigma}(s)}{\sigma(s)}$, $\gamma_1(s) := \frac{\dot{\alpha}(s)}{\alpha(s)}\sigma(s)^2 - \dot{\sigma}(s)\sigma(s)$, $\gamma_2(s) := \frac{\dot{\alpha}(s)}{\alpha(s)}$.*

*Proof.* For simplicity, without loss of generality we consider unconditional flows $v_s(x)$ and distributions $p_s(x)$.

We begin by showing that Eq. (2.3) is contained in $\mathcal{V}(p)$, reproducing the proof of Albergo et al. [2023], Proposition 2.6, using characteristic functions. The characteristic function $g(s, k)$ for $p_s(x)$ is given by:

$$g(s, k) := \mathbb{E}[e^{ik^\top(\alpha(s)X_0 + \sigma(s)X_1)}]$$

$$= \int e^{ik^\top x} p_s(x)dx$$

Taking the time derivative, by the Liebniz rule we have,

$$\int e^{ik^\top x} \partial_s p_s(x)dx = \partial_s g(s, k)$$

$$= \mathbb{E}[ik^\top(\dot{\alpha}(s)X_0 + \dot{\sigma}(s)X_1)e^{ik^\top(\alpha(s)X_0 + \sigma(s)X_1)}]$$

$$= ik^\top \int \mathbb{E}[\dot{\alpha}(s)X_0 + \dot{\sigma}(s)X_1 | X_s = x]e^{ik^\top x} p_s(x)dx$$

$$= ik^\top \int e^{ik^\top x} v_s(x)p_s(x)dx.$$

Note that for a differentiable scalar function $f : \mathbb{R} \to \mathbb{R}$ such that $\lim_{|x| \to \infty} f(x) = 0$, integration by parts yields the basic property of Fourier transforms,

$$\int e^{ikx} \frac{df}{dx}(x)dx = \lim_{L \to \infty} [f(x)e^{ikx}]_{-L}^L - \int f(x)\frac{d}{dx}e^{ikx}dx$$

$$= -ik \int e^{ikx} f(x)dx.$$

Applying this in the above, we have

$$ik^\top \int e^{ik^\top x} v_s(x) p_s(x) dx = \sum_j ik_j \int e^{ik_j} v_j(x,s) p_s(x) dx_j$$

$$= -\sum_j \int e^{ik_j} \frac{\partial}{\partial x}[v_j(x,s)p_s(x)] dx_j$$

$$= -\int e^{ik_j} \left( \sum_j \frac{\partial}{\partial x}[v_j(x,s)p_s(x)] \right) dx_j$$

$$= -\int e^{ik^\top x} \nabla \cdot (v_s(x)p_s(x)) dx.$$

We can conclude therefore that,

$$\partial_s p_s(x) + \nabla \cdot (v_s(x)p_s(x)) = 0,$$

meaning that $(v, p)$ constitute a normalizing flow per Eq. (A.2).

We now proceed to additionally show that Eq. (2.3) is the minimum-norm such solution. Consider any perturbation $\delta v$ such that $v + \delta v$ remains a solution of Eq. (A.2), i.e. $\nabla \cdot (\delta v_s(x) p_s) = 0$.

We begin by forming the Lagrangian:

$$\inf_v \sup_\lambda L(v, \lambda) = \int \|v\|^2 p_s(x) dx + \int \lambda(x)(\nabla \cdot (vp_s) + \partial_s p_s(x))(x) dx$$

We now apply the optimality condition and use integration by parts to obtain:

$$0 = L(v + \delta v, \lambda) - L(v, \lambda) - O(\|\delta v\|^2) \qquad \forall \delta v$$

$$= \int \delta v^\top v p_s(x) dx + \int \lambda(x) \nabla \cdot (\delta v p_s) dx \qquad \forall \delta v$$

$$= \int \delta v^\top v p_s(x) dx + \int p_s(x) \delta v^\top \nabla \lambda(x) dx \qquad \forall \delta v$$

$$= \int p_s(x) \delta v^\top [v + \nabla \lambda(x)] dx \qquad \forall \delta v$$

$$v = -\nabla \lambda(x).$$

This implies that a flow $v$ is optimal provided it satisfies the constraint $\partial_s p_s(x) + \nabla \cdot (vp_s(x)) = 0$ and there exists a $\lambda$ such that $v = -\nabla \lambda(s)$, i.e. if $v$ is conservative. Therefore all that remains is to show that $v$ is in fact conservative. We use Tweedie's formula to rewrite $v$:

$$v_s(x) = \mathbb{E}[\dot\alpha(s) X_0 + \dot\sigma(s) X_1 | X_s = x]$$

$$= \mathbb{E}[\dot\alpha(s) X_0 + \frac{\dot\sigma(s)}{\sigma(s)}(X_s - \alpha(s)X_0) | X_s = x, Z = z]$$

$$= \left( \dot\alpha(s) - \frac{\dot\sigma(s)}{\sigma(s)} \alpha(s) \right) \mathbb{E}[X_0 | X_s = x, Z = z] + \frac{\dot\sigma(s)}{\sigma(s)} x$$

$$= \left( \dot\alpha(s) - \frac{\dot\sigma(s)}{\sigma(s)} \alpha(s) \right) \frac{1}{\alpha(s)}(x + \sigma(s)^2 \nabla_x \log p_s(x|z)) + \frac{\dot\sigma(s)}{\sigma(s)} x$$

$$= \left( \frac{\dot\alpha(s)}{\alpha(s)} - \frac{\dot\sigma(s)}{\sigma(s)} \right)(x + \sigma(s)^2 \nabla_x \log p_s(x|z)) + \frac{\dot\sigma(s)}{\sigma(s)} x$$

$$= \left( \frac{\dot\alpha(s)}{\alpha(s)} \sigma(s)^2 - \dot\sigma(s)\sigma(s) \right) \nabla_x \log p_s(x) + \frac{\dot\alpha(s)}{\alpha(s)} x.$$

This shows both the equivalence of Eq. (2.3) and Eq. (2.4) and that Eq. (2.3) is conservative and hence, the ideal flow. □

## A.2 Proofs Regarding Forward and Reverse Processes (Appendix B)

**Proposition B.1** (Stochastic Generative Processes). *Given a conditional flow $(v, p)$ and conditioning value $z \in \mathcal{Z}$, let $\epsilon : [0, 1] \to \mathbb{R}_{\geq 0}$ be a time-dependent noise scale. Use $\{X_s^F\}_{s\in[0,1]}|Z$ and $\{X_s^B\}_{s\in[0,1]}|Z$ to denote the forward and reverse processes where $\mathrm{Law}(X_0^F|Z = z) = p_0(\cdot|z)$, $\mathrm{Law}(X_1^B|Z = z) = p_1(\cdot|z)$ and $X_s^F, X_{\hat{s}}^F$ are evolved according to,*

$$d(X_s^F|Z) = [v_s(X_s^F, Z) + \epsilon(s)\nabla_x \log p_s(X_s^F|Z)]ds + \sqrt{2\epsilon(s)}dB_s \qquad (B.1)$$

$$d(X_{\hat{s}}^R|Z) = [-v_{\hat{s}}(X_{\hat{s}}^B, Z) + \epsilon(\hat{s})\nabla_x \log p_{\hat{s}}(X_s^R|Z)]d\hat{s} + \sqrt{2\epsilon(s)}dB_{\hat{s}}. \qquad (B.2)$$

*where $\hat{s} := 1 - s$ and $B_s, B_{\hat{s}}$ are standard Brownian noise processes. In particular, $X_s^F, X_s^R$ these processes satisfy,*

$$\mathrm{Law}(X_s^F|Z = z) = \mathrm{Law}(X_s^R|Z = z) = p_s(\cdot|z).$$

*Proof.* For simplicity we only consider the following unconditional forward SDE.

$$dX_s^F = f(X_s^F, s)ds + \sqrt{2\epsilon(s)}dB_s \qquad (A.1)$$

Let $p_t(x)$ be the continuous density associated with (A.1) with $X_0 \sim p_0(x)$. The Fokker-Planck equation [Risken, 1996] yields,

$$\frac{\partial}{\partial s}p_s(x) = -\sum_{i=1}^{d_x}\frac{\partial}{\partial x_i}[f(x, s)p_s(x)] + \epsilon(t)\sum_{i=1}^{d_x}\sum_{j=1}^{d_x}\frac{\partial^2}{\partial x_i \partial x_j}[I \cdot p_s(x)]_{ij}$$

$$= -\sum_{i=1}^{d_x}\frac{\partial}{\partial x_i}[f(x, s)p_s(x)] + \epsilon(t)p_s(x)\sum_{i=1}^{d_x}\frac{\partial^2}{\partial x_i^2}[\log p_s(x)]$$

$$= -\nabla \cdot [f(x, s)p_s(x) - \epsilon(t)p_s(x)\nabla \log p_s(x)]$$

Letting $f(x, s) = v_s(x) + \epsilon(s)\nabla \log p_s(x) = 0$, we have

$$\frac{\partial}{\partial s}p_s = -\nabla \cdot (v_s(x)p_s(x)).$$

Which we can see is precisely the transport equation Eq. (A.2) for the $(v, p)$ flow. The reverse follows similarly:

$$\frac{\partial}{\partial \hat{s}}p_{\hat{s}} = \nabla_x \cdot (v_{\hat{s}}(x)p_{\hat{s}}(x)) = -\nabla_x \cdot (-v_{\hat{s}}(x)p_{\hat{s}}(x)).$$

Let $g(x, \hat{s}) := -v_{\hat{s}}(x) + \epsilon(\hat{s})\nabla \log p_{\hat{s}}(x)$. Then,

$$\frac{\partial}{\partial \hat{s}}p_{\hat{s}} = -\nabla \cdot (g(x, \hat{s})p_{\hat{s}}(x) - \epsilon(s)\nabla p_{\hat{s}}(x) \log p_{\hat{s}}(x))$$

Thus we can see that,

$$dX_{\hat{s}}^R = g(X_{\hat{s}}^R, \hat{s})d\hat{s} + \sqrt{2\epsilon(s)}dB_s.$$

$\square$

## A.3 Proofs for Section 3

### A.3.1 Proof of Theorem 1

The proof of Theorem 1 is a variant of the performance-difference lemma, adapted to the control of the solution to PDEs.

**Lemma A.2** (Finite-Horizon Deterministic Performance-Difference Lemma). *Consider states $x \in \mathcal{S}$ and actions $u \in \mathcal{A}$ and a continuous-time dynamical system $\dot{x}(t) = f_t(x(t), u(t))$ defined over*

$t \in [0, T]$ for some $T > 0$. Let $\pi(x, t)$ to denote a feedback policy $\pi : \mathcal{S} \times [0, T] \to \mathcal{A}$ and $\mu$ be any finite positive measure over $\mathcal{B}([0, T])$, the Borel $\sigma$-algebra on $[0, T]$. Define,

$$V_{t_1, t_2}^{\pi}(x) := \int_{t_1}^{t_2} r_s(x^{\pi}(s), \pi(x^{\pi}(s)))d\mu(s) \quad \text{where } \dot{x}^{\pi}(s) = f_s(x(s), \pi(x(s))), x^{\pi}(t_1) = x.$$

$$A_{t_1, t_2}^{\pi}(x, \pi') := \lim_{\epsilon \to 0} \frac{1}{\epsilon}(V_{t_1, t_2}^{\hat{\pi}_{t_1, \epsilon}}(x) - V_{t_1, t_2}^{\pi}(x)) \quad \text{where } \hat{\pi}_{t_1, \epsilon}(x, t) = \begin{cases} \pi'(x, t) & \text{if } t \leq t_1 + \epsilon \\ \pi(x, t) & \text{if } t > t_1 + \epsilon \end{cases}$$

for any $\mu$-integrable function $r_s(x^{\pi}(s), \pi(x^{\pi}(s)))$. Then, for any $t_1, t_2 \in [0, T], x \in \mathcal{S}$ and policies $\pi, \pi'$,

$$V_{t_1, t_2}^{\pi'}(x) - V_{t_1, t_2}^{\pi}(x) = \int_{t_1}^{t_2} A_t^{\pi}(x^{\pi'}(t), \pi')dt$$

*Proof.* Consider any $\epsilon \in (0, t_2 - t_1)$. Then,

$$\begin{aligned} V_{t_1, t_2}^{\pi'}(x) - V_{t_1, t_2}^{\pi}(x) &= V_{t_1, t_1 + \epsilon}^{\pi'}(x) + V_{t_1 + \epsilon, t_2}^{\pi'}(x^{\pi'}(t_1 + \epsilon)) - V_{t_1, t_1 + \epsilon}^{\pi}(x) - V_{t_1 + \epsilon, t_2}^{\pi}(x^{\pi}(t_1 + \epsilon)) \\ &= V_{t_1 + \epsilon, t_2}^{\pi'}(x^{\pi'}(t_1 + \epsilon)) - V_{t_1 + \epsilon, t_2}^{\pi}(x^{\pi'}(t_1 + \epsilon)) \\ &\quad + V_{t_1 + \epsilon, t_2}^{\pi}(x^{\pi'}(t_1 + \epsilon)) - V_{t_1 + \epsilon, t_2}^{\pi}(x^{\pi}(t_1 + \epsilon)) + V_{t_1, t_1 + \epsilon}^{\pi'}(x) - V_{t_1, t_1 + \epsilon}^{\pi}(x) \\ &= V_{t_1 + \epsilon, t_2}^{\pi'}(x^{\pi'}(t_1 + \epsilon)) - V_{t_1 + \epsilon, t_2}^{\pi}(x^{\pi'}(t_1 + \epsilon)) \\ &\quad + V_{t_1, t_2}^{\hat{\pi}_{t_1, \epsilon}}(x) - V_{t_1, t_2}^{\pi}(x) \end{aligned}$$

Choose $\epsilon := \frac{t_2 - t_1}{K}$ for some $K \geq 0$. Then, recursively applying the above identity yields,

$$V_{t_1, t_2}^{\pi'}(x) - V_{t_1, t_2}^{\pi}(x) = \sum_{k=0}^{K-1} V_{t_1 + k\epsilon, t_2}^{\hat{\pi}_{t_1 + k\epsilon, \epsilon}}(x^{\pi'}(t_1 + k\epsilon)) - V_{t_1 + k\epsilon, t_2}^{\pi}(x^{\pi'}(t_1 + k\epsilon))$$

Taking the limit $K \to \infty$,

$$\begin{aligned} V_{t_1, t_2}^{\pi'}(x) - V_{t_1, t_2}^{\pi}(x) &= \lim_{K \to \infty} \sum_{k=0}^{K-1} [A_t^{\pi}(x^{\pi'}(t_1 + k\epsilon), \pi'))\epsilon + o(\epsilon)] \\ &= \int_{t_1}^{t_2} A_t^{\pi}(x^{\pi'}(t), \pi')dt \end{aligned}$$

$\square$

**Notation.** We use $p_{t|s}^v$ to denote the solution to [Eq. (A.2)] with $v$ and initial condition $p_{s|s}^v = p_s$.

**Lemma A.3** (Diffusion Performance-Difference Lemma). *Let $r_s$ be some time-varying functional defined over $s \in [0, 1]$ which maps continuous densities to scalar values. For any finite positive measure $\mu$ over $\mathcal{B}([0, 1])$, define*

$$V_s^v(p_s) = \int_s^1 r_t(p_{t|s}^v)d\mu(t)$$

$$A_s^v(p_s, v') = \lim_{\epsilon \to 0} \frac{1}{\epsilon}\left[V_s^{w_{t|s}^{\epsilon}}(p_s) - V_s^v(p_s)\right]$$

*where $w_{t|s}^{\epsilon} = \begin{cases} v_t' \text{ if } t \leq s + \epsilon \\ v_t \text{ otherwise.} \end{cases}$ . Then the difference between $V_s^{v'}(p_s)$ and $V_s^v(p_s)$ can be written:*

$$V_s^{v'}(p_s) - V_s^v(p_s) = \int_s^1 A_t^v(p_{t|s}^{v'}, v')dt$$

*Proof.* This is just [Lemma A.2], where $\mathcal{S} \subset C^1(\mathcal{X}, \mathbb{R}_+)$ are densities over $\mathcal{X}$, actions $\mathcal{A}$ are maps $\mathcal{X} \to \mathbb{R}^{d_x}$ and policies $\pi$ are flows $v : [0, 1] \times \mathcal{X} \to \mathbb{R}^{d_x}$. $\square$

**Lemma A.4.** *Let $p_s, p_s'$ be densities and $v$ be a flow such that $(p_s, v), (p_s', v)$ satisfy Eq. (A.2). Then for any $\bar{p}_s$, $t \in [0, 1]$*

$$|\mathrm{TV}(p_s', \bar{p}_s) - \mathrm{TV}(p_s, \bar{p}_s)| \leq 2\mathrm{TV}(p_t, p_t')$$

*Proof.* Fix any $s, t \in [0, 1]$ and let $\alpha := \mathrm{TV}(p_t, p_t')$. Without loss of generality assume that $\mathrm{TV}(p_s', \hat{p}_s) \geq \mathrm{TV}(p_s, \hat{p}_s)$.

Decompose $p_t' = (1 - \alpha)p_t + \alpha \hat{p}_t$, where $\hat{p}_t$ is a signed density such that $\int |\hat{p}_t| dx = 1$ and $(\hat{p}_s, v)$ satisfy Eq. (A.2). Since Eq. (A.2) is linear in $p$, we can write $p_s' = (1 - \alpha)p_s + \alpha \hat{p}_s$ for all $s$. Thus,

$$\mathrm{TV}(p_s', \bar{p}_s) \leq (1 - \alpha)\mathrm{TV}(p_s, \bar{p}_s) + \alpha \mathrm{TV}(|\hat{p}_s|, \bar{p}_s).$$

Combining, we have that

$$\begin{aligned}|\mathrm{TV}(p_s', \bar{p}_s) - \mathrm{TV}(p_s, \bar{p}_s)| &\leq |(1 - \alpha)\mathrm{TV}(p_s, \bar{p}_s) + \alpha \mathrm{TV}(|\hat{p}_s|, \bar{p}_s) - \mathrm{TV}(p_s, \bar{p}_s)| \\ &\leq |-\alpha \mathrm{TV}(p_s, \bar{p}_s) + \alpha \mathrm{TV}(|\hat{p}_s|, \bar{p}_s)| \leq 2\alpha.\end{aligned}$$

$\square$

**Lemma A.5.** *Let $(p_s, v_s), (\hat{p}_s, \hat{v}_s)$ be pairs of solutions to Eq. (A.2). Let $(p_{s|t}, v_s)$ be a solution to Eq. (A.2) such that $p_{t|t} = \hat{p}_t$. Then, for any $t, s$,*

$$\mathrm{TV}(p_t, \hat{p}_t) \geq \frac{1}{2}(\mathrm{TV}(p_{s|t}, \hat{p}_s) - \mathrm{TV}(p_s, \hat{p}_s))$$

*Proof.* Applying Lemma A.4 using $p_s' = p_{s|t}, \bar{p}_s = \hat{p}_s$,

$$2\mathrm{TV}(p_t, p_{t|t}) \geq |\mathrm{TV}(p_{s|t}, p_s') - \mathrm{TV}(p_s, p_s')|.$$

Since we chose $p_{t|t} = \hat{p}_t$, this yields the desired statement

$$\begin{aligned}\mathrm{TV}(p_t, \hat{p}_t) &\geq \frac{1}{2}|\mathrm{TV}(p_{s|t}, \hat{p}_s) - \mathrm{TV}(p_s, \hat{p}_s)| \\ &\geq \frac{1}{2}(\mathrm{TV}(p_{s|t}, \hat{p}_s) - \mathrm{TV}(p_s, \hat{p}_s)).\end{aligned}$$

$\square$

**Theorem 1.** *Consider the setting of Definition 3.1, for a conditional flow $(v, p)$. For any probability measure $\mu$ over $[0, 1]$, the total variation distance $\mathrm{TV}(q, p) := \frac{1}{2}\int |p - q| \mathrm{d}x$ between $p_s$ and $p_s^{\mathrm{IMCF}}$ over $\mu$ is upper bounded by integrated schedule deviation:*

$$\int \mathrm{TV}(p_s(\cdot|z), p_s^{\mathrm{IMCF}}(\cdot|z))d\mu(s) \leq \int_0^1 \mathrm{SD}(v; z, t)\mathrm{d}t = \mathrm{SD}_{\mathrm{total}}(v, z).$$

*Moreover, if $|\partial_s^2 p_s^{\mathrm{IMCF}}|, |\partial_{st}^2 p_{s|t}^v|, |\partial_s^2 p_s| \leq M < \infty$, there exists constants $\epsilon_0 \in [0, 1/2], c \geq 0$ depending on $M$ such that, for all $0 < \epsilon \leq \epsilon_0$, $s \in [0, 1]$*

$$\sup_{t \in [s-\epsilon, s+\epsilon] \cap [0,1]} \mathrm{TV}(p_t(\cdot|z), p_t^{\mathrm{IMCF}}(\cdot|z)) \geq c\epsilon\mathrm{SD}(v; z, s). \tag{3.1}$$

*Proof.* We omit $z$ without loss of generality and consider the value function $V_s^v(p_s)$ given by

$$V_s^v(p_s) = \int \mathbb{1}\{t \geq s\} \cdot \mathrm{TV}(p_t^v, p_t^{\mathrm{IMCF}})d\mu(t)$$

The Diffusion Performance Difference Lemma Lemma A.3 gives the upper bound

$$\int \mathrm{TV}(p_t, p_t^{\mathrm{IMCF}})d\mu(t) = V_0^v(p_0) - V_0^{v^{\mathrm{IMCF}}}(p_0)$$

$$= -\int_0^1 A_s^v(p_s^{\mathrm{IMCF}}, v^{\mathrm{IMCF}})ds$$

Expanding $A_s^v(p_s^{\text{IMCF}}, v^{\text{IMCF}})$,

$$A_s^v(p_s^{\text{IMCF}}, v^{\text{IMCF}}) = \lim_{\epsilon \to 0} \frac{1}{\epsilon} \Big[ \int \mathbb{1}\{t \in [s, s+\epsilon]\} \cdot [-\text{TV}(p_{t|s}, p_t^{v^{\text{IMCF}}})] d\mu(t)$$

$$+ \int \mathbb{1}\{t \geq s+\epsilon\} \cdot [\text{TV}(p_{t|s+\epsilon}, p^{\text{IMCF}}) - \text{TV}(p_{t|s}, p_t^{\text{IMCF}})] d\mu(t) \Big]$$

Applying Lemma A.4, we can bound the integrand in the second integral by $\text{TV}(p_{s+\epsilon|s}, p_{s+\epsilon}^{\text{IMCF}})$

$$|A_s^v(p_s^{v^{\text{IMCF}}}, v^{\text{IMCF}})| \leq \lim_{\epsilon \to 0} \frac{1}{\epsilon} \Big[ \sup_{t \in [s, s+\epsilon]} 2\text{TV}(p_{t|s}, p_t^{\text{IMCF}}) \Big]$$

Taking a first order expansion, we have that for small $\epsilon$

$$\text{TV}(p_{s+\epsilon|s}^v, p_{s+\epsilon}^{\text{IMCF}}) = \frac{\epsilon}{2} \int \left| \left[ \frac{\partial p_{s|t}^v}{\partial s} \right]_{t=s} - \frac{\partial p_s^{\text{IMCF}}}{\partial s} \right| dx + o(\epsilon)$$

Thus $|A_s^v(p_s^{v^{\text{IMCF}}}, v^{\text{IMCF}})| \leq \text{SD}(v; z, t)$. Note that the proof also holds for the time-reversed direction since $p_1 = p_1^{\text{IMCF}}$. This yields the upper bound,

$$\int \text{TV}(p_t(\cdot|z), p_s^{\text{IMCF}}(\cdot|z)) d\mu(s) \leq \int_0^1 \text{SD}(v; z, t) \mathrm{d}t.$$

**Lower Bound**: For the lower bound, assuming that $|[\partial_s^2 p_{s|t}^v]_{t=s}|$, $|\partial_s[\partial_s p_{s|t}^v]_{t=s}|$, and $|\partial_s^2 p_t|$ all bounded by $M$, there exist constants $c > 0, \epsilon_0 \in [0, 1/2]$ depending only on $M$ such that for any, $s \in [0, 1], t, t' \in [s - \epsilon_0, s + \epsilon_0]$.

$$\text{TV}(p_{t|t'}, p_t^{\text{IMCF}}) \geq 3c\epsilon \int \left| \left[ \frac{\partial p_{t|s}}{\partial t} \right]_{t=s} - \frac{\partial p_s^{\text{IMCF}}}{\partial s} \right| dx = 3c\epsilon \text{SD}(v; z, s).$$

Fix any $\epsilon \in [0, \epsilon_0], s \in [0, 1 - \epsilon]$.

**Case 1**: $\text{TV}(p_s, p_s^{\text{IMCF}}) \leq c\epsilon \text{SD}(v, z, s)$. Pick any $\epsilon' \in [-\epsilon, \epsilon]$. By Lemma A.5,

$$\text{TV}(p_{s+\epsilon'}, p_{s+\epsilon'}^{\text{IMCF}}) \geq \frac{1}{2} [\text{TV}(p_{s|s+\epsilon'}, p_s^{\text{IMCF}}) - \text{TV}(p_s, p_s^{\text{IMCF}})]$$

$$\geq \frac{1}{2} [3c\epsilon' \text{SD}(v; z, s) - c\epsilon \text{SD}(v; z, s)]$$

$$\geq c\epsilon' \text{SD}(v; z, s).$$

Thus,

$$\sup_{\epsilon' \in [0, \epsilon_0]} \text{TV}(p_{s+\epsilon}, p_{s+\epsilon}^{\text{IMCF}}) \geq c\epsilon_0 \text{SD}(v; z, s).$$

**Case 2**: $\text{TV}(p_s, p_s^{\text{IMCF}}) \geq c\epsilon \text{SD}(v, z, s)$. In this case we trivially have

$$\sup_{\epsilon' \in [0, \epsilon]} \text{TV}(p_{s+\epsilon}, p_{s+\epsilon}^{\text{IMCF}}) \geq c\epsilon \text{SD}(v; z, s).$$

Note that since the argument is symmetric with respect to time, for $s \geq 1 - \epsilon_0$, we can consider $\epsilon \in [-\epsilon, 0]$. Thus for any $s \in [0, 1], \epsilon \in [0, \epsilon_0]$

$$\sup_{t \in [s-\epsilon, s+\epsilon]} \text{TV}(p_t, p_t^{\text{IMCF}}) \geq c\epsilon \text{SD}(v; z, s)$$

$\square$

### A.3.2  Proof of Proposition 3.1

**Proposition 3.1.** *It holds that* $\text{SD}(v; z, s) = \mathbb{E}_{p_s^{\text{IMCF}}}[|\nabla \cdot (v_s - v_s^{\text{IMCF}}) + (v_s - v_s^{\text{IMCF}}) \cdot \nabla \log p_s^{\text{IMCF}}|]$.

*Proof.* An equivalent condition for the pair $(p, v)$ to constitute a conditional flow is that it satisfies the differential transport equation [Albergo et al., 2023],

$$\partial_t p_t(x|z) + \nabla_x \cdot [v_t(x, z)p_t(x|z)] = 0. \tag{A.2}$$

This permits us to compute

$$\begin{aligned}
\text{SD}(v; z, s) &:= \int_{\mathcal{X}} \left| \left[ \frac{\partial p_{s|t}^s}{\partial s} \right]_{t=s} - \frac{\partial p_s^{\text{IMCF}}}{\partial s} \right| dx = \int_{\mathcal{X}} |\nabla \cdot [v_s p_s^{\text{IMCF}}] - \nabla \cdot [v_s^{\text{IMCF}} p_s^{\text{IMCF}}]| \, dx \\
&= \int_{\mathcal{X}} |\nabla \cdot (v_s - v_s^{\text{IMCF}}) + (v_s - v_s^{\text{IMCF}}) \cdot \nabla \log p_s^{\text{IMCF}}| \cdot p_s^{\text{IMCF}}(x) dx \\
&= \mathbb{E}_{p_s^{\text{IMCF}}}[|\nabla \cdot (v_t - v_t^{\text{IMCF}}) + (v_t - v_t^{\text{IMCF}}) \cdot \nabla \log p_s^{\text{IMCF}}|]. \tag{A.3}
\end{aligned}$$

$\square$

## A.4 Appendix for Extrapolation Behavior (Section 4)

### A.4.1 Proof of Theorem 2

**Theorem 2** (Discrete-support Smooth Interpolant). *Let $Z$ be supported on a finite set $S_z = \{z^{(i)}\}_{i=1}^N \subset \mathbb{R}$ for distinct $z^{(1)} < \ldots < z^{(N)}$, ordered without loss of generality. For each $z^{(i)} \in S_z$, let $v_s^\star(x, z^{(i)}) := \mathbb{E}[\dot\alpha(s)X_0 + \dot\sigma(s)X_1|Z = z^{(i)}]$. Then, there are piecewise cubic polynomials $p^{(i)}(z)$, with pieces defined by the intervals $[z^{(j)}, z^{(j+1)}]$ such that*

$$v_s(x, z) = \underset{v \in \arg\inf_v L[v]}{\arg\inf} \int \|\nabla_z^2 v\|_F^2 \mathrm{d}x = \sum_{z^{(i)} \in S_z} p^{(i)}(z) \cdot v_s^\star(x, z^{(i)}).$$

*In the case where $|S_z| = 2$, $p^{(i)}(z)$ are linear functions.*

*Proof.* This is the cubic spline interpolation, which traces back to the classic work of Kochanek and Bartels [1984], but we apply here in function space. The Euler-Lagrange equation for the functional $J(f) = \int_{x_1}^{x_2} |f''(x)|^2 \, \mathrm{d}x$ is:

$$\frac{\mathrm{d}^4}{\mathrm{d}x^4} f(x) \equiv 0 \text{ when } x \in (x_1, x_2).$$

Thus, applied to our setting, we have,

$$\frac{\partial^4}{\partial z^4} v_s(x, z) \equiv 0 \quad \forall z \in (z^{(i)}, z^{(i+1)}), s \in [0, 1], x \in \mathcal{X},$$

for each $i = 0, 2, \ldots, N$, where for convenience we let $z^{(0)} = -\infty$, $z^{(N+1)} = +\infty$. This means that on each interval $z \in [z^{(i)}, z^{(i+1)}], i \in \{1, \ldots, N-1\}$, the interpolator $v_s(x, z)$ can be written as a piecewise cubic polynomial in $z$ of the form

$$v_s(x, z) = a_s^{(i)}(x) + b_s^{(i)}(x)(z - z^{(i)}) + c_s^{(i)}(x)(z - z^{(i)})^2 + d_s^{(i)}(x)(z - z^{(i)})^3$$

where for $z \in (-\infty, z^{(1)}]$ we let $v_s(x, z) = v_s(x, z^{(1)}) + b_s^{(1)}(x) \cdot (z - z^{(1)})$ and similarly for the interval $z \in [z^{(N)}, \infty)$, we have $v_s(x, z) = v_s(x, z^{(N)}) + b_s^{(N)}(x) \cdot (z - z^{(N)})$.

Let $\Delta z_i = z^{(i+1)} - z^{(i)}$ and using $a^{(i)}, b^{(i)}, c^{(i)}, d^{(i)}$ as shorthand for $a_s^{(i)}(x), b_s^{(i)}(x), c_s^{(i)}(x), d_s^{(i)}(x)$, we have the following boundary conditions for the endpoints:

$$a^{(i)} = v_s^\star(x, z^{(i)}) \quad \forall i \in \{1, \ldots, N-1\}$$

$$a^{(i)} + b^{(i)}\Delta z_i + c^{(i)}\Delta z_i^2 + d^{(i)}\Delta z_i^3 = v_s^\star(x, z^{(i+1)}) \quad \forall i \in \{1, \ldots, N-1\}$$

and additionally, with boundary conditions to ensure the first and second derivatives match between the different pieces

$$b^{(i)} + 2c^{(i)}\Delta z_i + 3d^{(i)}\Delta z_i^2 = b^{(i+1)} \quad \forall i \in \{1, \ldots, N-2\}$$

$$2c^{(i)} + 6d^{(i)}\Delta z_i = 2c^{(i+1)} \quad \forall i \in \{1, \ldots, N-2\}$$

and we additionally constrain the second derivatives at the endpoint to be zero so that $c^{(1)} = 0$, $2c^{(i)} + 6d^{(i)}\Delta z_i = 0$. $\square$

This yields a linear system with $4(N-1)$ unknowns and equations. In fact, Bartels et al. [1995] shows that this system is guaranteed to be linearly independent.

Thus, we can write $a^{(i)}, b^{(i)}, c^{(i)}, d^{(i)}$'s as a linear combination of the $v^\star(x, z^{(i)})$'s. Therefore there exist piecewise polynomials $p^{(i)}(x)$ such that,

$$v(x, z) = \sum_{i=1}^{N} p^{(i)}(z) v^\star(x, z^{(i)}).$$

### A.4.2 Proof of Theorem 3

**Lemma A.6.**

$$L[v] = \mathbb{E}_Z \mathbb{E}_{X_s} \|v_s(X_s, Z) - v_s^\star(X_s, Z)\|^2 = \int_0^1 \int_{\mathcal{Z}} \int_{\mathcal{X}} \|v_s(x, z) - v_s^\star(x, z)\|^2 p_s^{\mathrm{IMCF}}(x, z) \mathrm{d}x \mathrm{d}z \mathrm{d}s \tag{A.4}$$

**Theorem 3.** *Fix some diffusion schedule $(\alpha, \sigma)$ and let $v_s^\star(x, z)$ be the IMCF flow associated with $p^\star(x|z)$, i.e. the minimizer of Eq. (2.2). Assume that $p^\star(z)$ is a uniform density over some set $S$, i.e. $p^\star(z) = c \cdot 1_S$ for some $c > 0$ and where $1_S$ the characteristic function of $S$. Then the minimizer to $L_\lambda[v]$ for any $z \in S$ is given by,*

$$v_s(x, z) = \int_{\xi, z' \in S} \frac{e^{2\pi i \xi(z - z')}}{1 + \lambda \|\xi\|^4} v^\star(x, z') \mathrm{d}z' \mathrm{d}\xi. \tag{4.2}$$

*Proof.* In light of Lemma A.6,

$$L_\lambda[v] = \int_0^1 (\|v_s(x, z) - v_s^\star(x, z)\|^2 + \lambda \|\nabla_z^2 v_s(x, z)\|^2) p_s^{\mathrm{IMCF}}(x, z) \mathrm{d}s. \tag{A.5}$$

Notice that the loss decouples across $s \in [0, 1]$ and $x \in \mathcal{X}$. Hence, let us consider the optimization problem for $s$ fixed. Set $\omega(z) = p_s^{\mathrm{IMCF}}(x, z)$, $f(z) = v_s(x, z)$ and $f^\star(z) = v_s^\star(x, z)$. Then the corresponding optimization for $s, x$ fixed becomes

$$\int_z (\|f(z) - f^\star(z)\|^2 + \lambda \|\nabla_z^2 f\|^2) \omega(z) \mathrm{d}z. \tag{A.6}$$

Since $\omega(z) = c \cdot 1_S$ is uniform, the Euler-Lagrange equation yields that for $z \in S$,

$$2c(f(z) - f^\star(z)) + 2c\lambda \sum_{i,j}^{n} \frac{\partial^4}{\partial^2 z_i \partial^2 z_j} = 0 \tag{A.7}$$

Thus, equivalently, for all $z \in \mathcal{Z}$,

$$f(z) \cdot 1_S + \lambda \sum_{i,j}^{n} \frac{\partial^4}{\partial z_i^2 \partial z_j^2} [f(z) \cdot 1_S] = f^\star(z) \cdot 1_S$$

We may therefore apply a Fourier transform to obtain

$$F(z) + \|\xi\|_2^4 F(z) = F^\star(z), \tag{A.8}$$

where $F(z)$ and $F^\star(z)$ denote the Fourier transforms of $f \cdot 1_S$ and $f^\star \cdot 1_S$ respectively, and where we invoke the conversion between differentiation and multiplication under the Fourier transform. Solving for $F(z)$, we have

$$F(z) = \frac{1}{1 + \lambda \|\xi\|_2^4} F^\star(z) = \frac{1}{1 + \lambda \|\xi\|_2^4} \int_S f^\star(z) e^{-2\pi i \xi z'} \mathrm{d}z' \tag{A.9}$$

Inverting gives, for any $z \in S$,

$$f(z) = \int e^{2\pi i \xi z} \frac{1}{1 + \lambda \|\xi\|_2^4} \int_S f^\star(z) e^{-2\pi i \xi z'} \mathrm{d}z' \tag{A.10}$$

$$= \int_{\xi, z' \in S} \frac{1}{1 + \lambda \|\xi\|_2^4} e^{2\pi i \xi(z - z')} f^\star(z') \mathrm{d}z' \mathrm{d}xi' \tag{A.11}$$

$\square$

# B  Sampling Algorithms and Schedule Deviation

The sampling process Definition 2.1 can equivalently be described using the following stochastic differential equation (see Appendix A.2 for proof):

**Proposition B.1** (Stochastic Generative Processes). *Given a conditional flow $(v, p)$ and conditioning value $z \in \mathcal{Z}$, let $\epsilon : [0, 1] \to \mathbb{R}_{\geq 0}$ be a time-dependent noise scale. Use $\{X_s^F\}_{s \in [0,1]} | Z$ and $\{X_s^B\}_{s \in [0,1]} | Z$ to denote the forward and reverse processes where $\mathrm{Law}(X_0^F | Z = z) = p_0(\cdot | z)$, $\mathrm{Law}(X_1^B | Z = z) = p_1(\cdot | z)$ and $X_s^F, X_{\hat{s}}^B$ are evolved according to,*

$$d(X_s^F | Z) = [v_s(X_s^F, Z) + \epsilon(s) \nabla_x \log p_s(X_s^F | Z)] ds + \sqrt{2\epsilon(s)} dB_s \qquad (B.1)$$

$$d(X_{\hat{s}}^R | Z) = [-v_{\hat{s}}(X_{\hat{s}}^B, Z) + \epsilon(\hat{s}) \nabla_x \log p_{\hat{s}}(X_s^R | Z)] d\hat{s} + \sqrt{2\epsilon(\hat{s})} dB_{\hat{s}}. \qquad (B.2)$$

*where $\hat{s} := 1 - s$ and $B_s, B_{\hat{s}}$ are standard Brownian noise processes. In particular, $X_s^F, X_s^R$ these processes satisfy,*

$$\mathrm{Law}(X_s^F | Z = z) = \mathrm{Law}(X_s^R | Z = z) = p_s(\cdot | z).$$

**Sampling Algorithms with IMCF Flows.** Given Corollary B.1, the continuous-time analogous of the sampling algorithms we chiefly consider (DDPM [Ho et al., 2020], DDIM [Song et al., 2020a], GE [Permenter and Yuan, 2023]). In particular, we note that DDPM/DDIM thus should sample from the equivalent distributions (in the continuous-time limit) under the assumption that the learned flow $v$ is IMCF:

**Corollary B.1** (IMCF-based generation). *Given an $(\alpha, \sigma)$-IMCF flow $(p, v)$, for any $\epsilon : [0, 1] \to \mathbb{R}_+$, using Eq. (2.4) the forward and reverse processes Eq. (B.1), and Eq. (B.2) can be written as,*

$$d(X_s^F | Z) = \left[ \left( 1 + \frac{\epsilon(s)}{\gamma_1(s)} \right) v_s(X_s^F, Z) + \frac{\gamma_2(s)}{\gamma_1(s)} X_s^F \right] ds + \sqrt{2\epsilon(s)} dB_s, \qquad (B.3)$$

$$d(X_{\hat{s}}^R | Z) = \left[ \left( \frac{\epsilon(\hat{s})}{\gamma_1(\hat{s})} - 1 \right) v_{\hat{s}}(X_{\hat{s}}^R, Z) + \frac{\gamma_2(\hat{s})}{\gamma_1(\hat{s})} X_{\hat{s}}^R \right] d\hat{s} + \sqrt{2\epsilon(\hat{s})} dB_{\hat{s}}, \qquad (B.4)$$

*where $\gamma_1(s) := \frac{\dot{\alpha}(s)}{\alpha(s)} \sigma(s)^2 - \dot{\sigma}(s)\sigma(s)$ and $\gamma_2(s) := \frac{\dot{\alpha}(s)}{\alpha(s)}$.*

**Example B.1** (DDPM [Ho et al., 2020]). The SDE-analogue of the DDPM sampling algorithm corresponds to the choice $\epsilon(s) = -\gamma_1(s)$ (note that $\gamma_1(s), \gamma_2(s) \leq 0$), making the forward process independent of $v$ and thus a purely Ornstein-Uhlenbeck process and the reverse process simplifies to,

$$d(X_{\hat{s}}^R | Z) = \left[ -2v_{\hat{s}}(X_{\hat{s}}^R, Z) + \frac{\gamma_2(\hat{s})}{\gamma_1(\hat{s})} X_{\hat{s}}^R \right] d\hat{s} + \sqrt{2\epsilon(\hat{s})} dB_{\hat{s}}. \qquad (B.5)$$

**Example B.2** (DDIM [Song et al., 2020a]). The DDIM algorithm strictly generalizes DDPM and technically allows for any choice of $\epsilon(s) \geq 0$. In practice (e.g. Karras et al. [2022], Chi et al. [2023]) DDIM is used in a "noiseless" fashion with $\epsilon = 0$, in which case the reverse process simply becomes the regular flow ODE,

$$d(X_{\hat{s}} | Z) = -v_{\hat{s}}(X_{\hat{s}}^R, Z) d\hat{s}$$

**Example B.3** (Gradient Estimation (GE) [Permenter and Yuan, 2023]). The Gradient Estimation algorithm is a variant of DDIM which introduces a correction term based on the previous estimate of $v$, i.e. it uses the filtered flow $\bar{v}(x_s | z) = \mu v(x_s | z) + (1 - \mu) v(x_{s + \delta s} | z)$ where $\delta s$ is the discretization interval of the SDE. Note that in the continuous time limit where $\delta s \to 0$, we recover the standard DDIM. We use $\mu = 2$ for the experiments presented here.

## B.1  Schedule Deviation Emerges from Linear Interpolation

To examine how schedule deviation can emerge from linear interpolation, we consider combining normal distributions with differing means and variances, respectively:

**Lemma B.2.** *Consider two scalar normal distributions, $p^{(1)}(x) = \mathcal{N}(\mu_1, \bar{\sigma}^2)$ and $p^{(2)}(x) = \mathcal{N}(\mu_2, \bar{\sigma}^2)$. For a given a Diffuion Schedule $(\sigma, \alpha)$, the associated score functions of the distributions at time $s$ are,*

$$\nabla \log p_s^{(1)}(x) = -\frac{x - \alpha(s)\mu_1}{\bar{\sigma}^2 + \sigma^2(s)}, \quad \nabla \log p_s^{(2)}(x) = -\frac{x - \alpha(s)\mu_2}{\bar{\sigma}^2 + \sigma^2(s)}.$$

*and, for any $c \in \mathbb{R}$, the combined score function $c\nabla \log p_s^{(1)}(x) + (1-c)\nabla \log p_s^{(2)}(x)$ is also consistent with the Diffusion Schedule $(\sigma, \alpha)$:*

$$c\nabla \log p_s^{(1)}(x) + (1-c)\nabla \log p_s^{(2)}(x) = -\frac{x - \alpha(s)(c\mu_1 + (1-c)\mu_2)}{\bar{\sigma}^2 + \sigma^2(s)}.$$

*We can recognize this as the score function for $p(x) = \mathcal{N}(c\mu_1 + (1-c)\mu_2, \bar{\sigma})$, interpolated in accordance with the Diffusion Schedule $(\sigma, \alpha)$.*

*Proof.* The proof follows by straightforward substitution. $\qquad\square$

**Lemma B.3.** *Consider two scalar normal distributions, $p^{(1)}(x) = \mathcal{N}(0, \bar{\sigma}^2)$ and $p^{(2)}(x) = \mathcal{N}(0, k^2\bar{\sigma}^2)$ for $k \geq 0, k \neq 1$. Then, given a Diffusion Schedule $(\sigma, \alpha)$, the associated score functions of the distributions at time $s$ are,*

$$\nabla \log p_s^{(1)}(x) = -\frac{x}{\bar{\sigma}^2\alpha^2(s) + \sigma^2(s)} = -\frac{1}{\alpha^2(s)} \cdot \frac{x}{\bar{\sigma}^2 + \hat{\sigma}^2(s)},$$

$$\nabla \log p_s^{(2)}(x) = -\frac{x}{k^2\bar{\sigma}^2\alpha^2(s) + \sigma^2(s)} = -\frac{1}{\alpha^2(s)} \cdot \frac{x}{k^2\bar{\sigma}^2 + \hat{\sigma}^2(s)}.$$

*where $\hat{\sigma}(s) = \frac{\sigma(s)}{\alpha(s)}$. Then for any $c \in \mathbb{R} \setminus \{0, 1\}$, the linear interpolation of the score function is given by,*

$$c\nabla \log p_s^{(1)}(x) + (1-c)\nabla \log p_s^{(2)}(x) = -\frac{1}{\alpha^2(s)} \cdot \frac{x}{\bar{\sigma}^2 + \bar{\sigma}^2(1-c)(k^2-1)\beta_{c,k}(s) + \hat{\sigma}^2(s)}.$$

*where $\beta_{c,k}(s) = \frac{\bar{\sigma}^2 + \sigma^2(s)}{(1+c(k^2-1))\bar{\sigma}^2 + \sigma^2(s)}$.*

*Proof.*

$$
\begin{aligned}
c\nabla \log p_s^{(1)}(x) + (1-c)\nabla \log p_s^{(2)}(x) &= -\frac{1}{\alpha^2(s)} \cdot \left( \frac{(1-c)(\bar{\sigma}^2 + \hat{\sigma}^2(s)) + c(k^2\bar{\sigma}^2 + \hat{\sigma}^2(s))}{(k^2\bar{\sigma}^2 + \hat{\sigma}^2(s))(\bar{\sigma}^2 + \hat{\sigma}^2(s))} \right) x \\
&= -\frac{1}{\alpha^2(s)} \cdot \left( \frac{(1-c+ck^2)\bar{\sigma}^2 + \hat{\sigma}^2(s)}{(k^2\bar{\sigma}^2 + \hat{\sigma}^2(s))(\bar{\sigma}^2 + \hat{\sigma}^2(s))} \right) x \\
&= -\frac{1}{\alpha^2(s)} \cdot \left( \frac{(1-c+ck^2)\bar{\sigma}^2 + \hat{\sigma}^2(s)}{k^2\bar{\sigma}^4 + (1+k^2)\bar{\sigma}^2\hat{\sigma}^2(s) + \hat{\sigma}^4(s)} \right) x \\
&= -\frac{1}{\alpha^2(s)} \cdot \frac{x}{\frac{k^2\bar{\sigma}^4 + (1+k^2)\bar{\sigma}^2\hat{\sigma}^2(s) + \hat{\sigma}^4(s)}{(c+(1-c)k^2)\bar{\sigma}^2 + \hat{\sigma}^2(s)}} \\
&= -\frac{1}{\alpha^2(s)} \cdot \frac{x}{\frac{k^2\bar{\sigma}^4 + (c+(1-c)k^2)\bar{\sigma}^2\hat{\sigma}^2(s) + (1-c+ck^2)\bar{\sigma}^2\hat{\sigma}^2(s) + \hat{\sigma}^4(s)}{(1-c+ck^2)\bar{\sigma}^2 + \hat{\sigma}^2(s)}} \\
&= -\frac{1}{\alpha^2(s)} \cdot \frac{x}{\frac{k^2\bar{\sigma}^4 + (c+(1-c)k^2)\bar{\sigma}^2\hat{\sigma}^2(s)}{(1-c+ck^2)\bar{\sigma}^2 + \hat{\sigma}^2(s)} + \hat{\sigma}^2(s)}.
\end{aligned}
$$

Letting $\bar{\sigma}_c^2(s) := \frac{k^2\bar{\sigma}^4 + (c+(1-c)k^2)\bar{\sigma}^2\sigma^2(s)}{(1-c+ck^2)\bar{\sigma}^2 + \sigma^2(s)}$, we can see that the linear combination is consistent with the Diffusion Schedule $(\sigma, \alpha)$ only if $\hat{\sigma}_c^2(s)$ is independent of $s$.

$$
\begin{aligned}
\bar{\sigma}_c^2(s) &= \frac{k^2\bar{\sigma}^4 + (c+(1-c)k^2)\bar{\sigma}^2\sigma^2(s)}{(1-c+ck^2)\bar{\sigma}^2 + \sigma^2(s)} \\
&= \bar{\sigma}^2 \left( \frac{k^2\bar{\sigma}^2 + (c+(1-c)k^2)\sigma^2(s)}{(1-c+ck^2)\bar{\sigma}^2 + \sigma^2(s)} \right) \\
&= \bar{\sigma}^2 \left( \frac{(1-c+ck^2)\bar{\sigma}^2 + \hat{\sigma}^2(s) + (c-1+(1-c)k^2)\bar{\sigma}^2 + (c-1+(1-c)k^2)\hat{\sigma}^2(s)}{(1-c+ck^2)\bar{\sigma}^2 + \hat{\sigma}^2(s)} \right) \\
&= \bar{\sigma}^2 + \bar{\sigma}^2 \left( \frac{(c-1+(1-c)k^2)\bar{\sigma}^2 + (c-1+(1-c)k^2)\hat{\sigma}^2(s)}{(1-c+ck^2)\bar{\sigma}^2 + \hat{\sigma}^2(s)} \right) \\
&= \bar{\sigma}^2 + \bar{\sigma}^2(1-c)(k^2-1) \left( \frac{\bar{\sigma}^2 + \hat{\sigma}^2(s)}{(1+c(k^2-1))\bar{\sigma}^2 + \hat{\sigma}^2(s)} \right).
\end{aligned}
$$

Thus,

$$
\begin{aligned}
&c\nabla \log p_s^{(1)}(x) + (1-c)\nabla \log p_s^{(2)}(x) \\
&= -\frac{1}{\alpha^2(s)} \cdot \frac{x}{\bar{\sigma}^2 + \bar{\sigma}^2(1-c)(k^2-1)\left(\frac{\bar{\sigma}^2+\sigma^2(s)}{(1+c(k^2-1))\bar{\sigma}^2+\sigma^2(s)}\right) + \hat{\sigma}^2(s)}.
\end{aligned}
$$

$\square$

We can see that for any fixed $s$, the combined $c\nabla \log p_s^{(1)} + (1-c)\nabla \log p_s^{(2)}$ resembles the score function of a normal distribution, but the dependence of $\beta_{c,k}(s)$ on $s$ in the denominator indicates that it is noised according to a different schedule from $(\sigma, \alpha)$.

Taken together Lemma B.2 and Lemma B.3 suggest that schedule inconsistency can arise through differences in variance, but not simple transformations such as translation.

## C   Experiment Details

All experiments were performed using a cluster of 4 NVIDIA A100 GPUs and took approximately 100 GPU/hrs of compute to train and evaluate all visualized experiments. Combined, all experiments took approximately one week worth of GPU/hours to train and evaluate.

### C.1   Conditional CelebA Experiments

We additionally consider an ablation on the CelebA [Liu et al., 2015] dataset, using a t-SNE of the 40-dimensional conditional attribute space to conditionally generate 64x64 images, as described in Appendix C.5. We use training datasets of size $N \in \{50000, 100000, 160000\}$ for our experiments.

Notably, for these experiments *we omit the divergence term* from the Schedule Deviation computation (i.e. setting $r_1 = 0$ in Algorithm 1). This is principally for computational reasons–we do not have enough memory to evaluate the divergence.

In Figure 8 we visualize the corresponding conditioning space, highlighting 3 different conditioning attributes (Male/Female, Young/Not Young, Smiling/Not Smiling), as well as the OT distance/Schedule Deviation over the space.

Similar to the MNIST, Fashion-MNIST and trajectory datasets, we observe little change in Schedule Deviation based on the amount of training data used. However, unlike the MNIST, Fashion-MNIST and Maze datasets, we observe no correlation between Schedule Deviation and the computed Optimal Transport distances, despite similar overall structure in the OT distances and Schedule Deviation as visualized in Figure 9.

We hypothesize the lack of correlation is related to the relatively uniform coverage of the dataset over the conditioning space and the much higher dimension for the generated space $\mathcal{X}$. Notably, we also observe very little variance in the OT cost. By contrast, MNIST, Fashion-MNIST, and the Maze experiments all have very non-uniform coverage over the conditioning space and exhibit much lower noise (i.e. smoother heatmaps) for the Schedule Deviation estimates.

Overall, we believe these experiments are somewhat inconclusive, partially due to the modified methodology (using $r_1 = 0$) and the lack of significant structure either for the OT distances or the Schedule Deviation.

### C.2   Measuring Schedule Deviation

For simplicity we consider Diffusion Schedules (Definition 2.3) which are purely noising, i.e where $\alpha(s) = 1$. In practice we assume that $s \in (0, 1)$, so any such schedule can easily be normalized into the standard form via the transformation $x \to \frac{x}{1+\sigma(s)}$.

Thus, in this simplified setting, the loss simply becomes,

$$
L_v := \mathbb{E}_{x,\epsilon}[\|v_s(x + \sigma(s)\epsilon, z) - \dot{\sigma}(s)\epsilon\}\|^2] \tag{C.1}
$$

where $x \sim p^\star(x)$ and $\epsilon \sim \mathcal{N}(0, I)$. Under this framework, per Eq. (2.4), the minimizer to Eq. (C.1) can be written $v_s^{\text{IMCF}}(x, z) := -\dot{\sigma}(s)\sigma(s)\nabla_x \log p_s^{\text{IMCF}}(x, z)$. For convenience, we use the "$\epsilon$-parameterization" of the flow introduced in [Ho et al., 2020], wherein $v_s(x, z) = -\dot{\sigma}(s)\epsilon_s(x, z)$

**Schedule Deviation Estimator** Thus, we can estimate the schedule deviation using,

$$
\begin{aligned}
\text{SD}(v; z, s) &= \mathbb{E}_{p_s^{\text{IMCF}}}[|\nabla \cdot (v_s - v_s^{\text{IMCF}}) + (v_s - v_s^{\text{IMCF}}) \cdot \nabla \log p_s^{\text{IMCF}}|] \\
&= \dot{\sigma}(s)\mathbb{E}_{p_s^{\text{IMCF}}}[|\nabla \cdot (\epsilon_s - \epsilon_s^{\text{IMCF}}) + (\epsilon_s^{\text{IMCF}} - \epsilon_s) \cdot \sigma^{-1}(s)\epsilon_s^{\text{IMCF}}|] \\
&= \mathbb{E}_{p_s^{\text{IMCF}}}\left[\left|\dot{\sigma}(s)\nabla \cdot (\epsilon_s - \epsilon_s^{\text{IMCF}}) + \frac{\dot{\sigma}(s)}{\sigma(s)}(\epsilon_s^{\text{IMCF}} - \epsilon_s) \cdot \epsilon_s^{\text{IMCF}}\right|\right].
\end{aligned}
$$

**Empirical Estimation of $\epsilon_s^{\text{IMCF}}$.** The quantity $\epsilon_s$ above is directly parameterized by the neural network and we can sample from $x_s \sim p_s^{\text{IMCF}}(x_s|z)$ by generating samples from $p_0(x|z)$ (using the designated sampling algorithm) and then noising to time $s$ using the forward process. Estimating $\epsilon_s^{\text{IMCF}}$ is less straightforward. Here we use that for $N$ samples $\{x^{(i)}\}_{i=1}^N$ from $p_0(x|z)$, we can approximate $\nabla \log p_s^{\text{IMCF}}(x|z)$ and therefore $\epsilon^{\text{IMCF}}$ (using $\mathcal{N}(\cdot; \mu, \sigma^{\in})$ to denote the Gaussian PDF with mean $\mu$ and variance $\sigma^2$):

$$
\begin{aligned}
\epsilon_s^{\text{IMCF}}(x, z) &\approx \sigma(s)\nabla \log\left(\frac{1}{N}\sum_{i=1}^N \mathcal{N}(x; x^{(i)}, \sigma^2(s))\right) \\
&= \sigma(s)\frac{1}{\sum_{i=j}^N \exp\left(-\frac{\|x-x^{(j)}\|_2^2}{2\sigma^2(s)}\right)}\sum_{i=1}^N \exp\left(-\frac{\|x-x^{(i)}\|_2^2}{2\sigma^2(s)}\right)\left(\frac{x-x^{(i)}}{\sigma^2(s)}\right) \\
&= \sum_{i=1}^N \left[\frac{\exp\left(-\frac{\|x-x^{(i)}\|_2^2}{2\sigma^2(s)}\right)}{\sum_{j=1}^N \exp\left(-\frac{\|x-x^{(j)}\|_2^2}{2\sigma^2(s)}\right)}\right]\left(\frac{x-x^{(i)}}{\sigma(s)}\right).
\end{aligned}
$$

In the MNIST, Fashion-MNIST, CelebA, and Trajectory experiments we use $N = 128$ whereas for the toy experiments in Section 4 we use $N = 2000$.

**Empirical Estimation of $\nabla \cdot (\epsilon_s - \epsilon_s^{\text{IMCF}})$.** Computing $\nabla \cdot (\epsilon_s - \epsilon_s^{\text{IMCF}})$ requires computing the divergence of a neural network, i.e. the trace of the Jacobian. For a function $f : \mathbb{R}^n \to \mathbb{R}^n$, computing the divergence requires $O(n)$ Jacobian-vector products, i.e. it is as computationally expensive as materializing the full $n \times n$ Jacobian. Thus, we consider instead a randomized approximation to the divergence wherein we randomly sample a standard basis vector (i.e. an element of the diagonal of the Jacobian) to use as an estimate of the true divergence for each sample $x_s^{(i)} \sim p_s^{\text{IMCF}}(x_s|z)$.

**Schedule Deviation with Log-Linear Schedules.** In practice (see Appendix C.5) we use a log-linear noise schedule $\sigma(t)$, where $\sigma(s) = c_1 e^{c_2 s}$ and $c_1, c_2$ are chosen based on the desired $\sigma(0)$ and $\sigma(1)$, i.e. $c_1 = \sigma(0), c_2 = \log(\sigma(1)/\sigma(0))$. Thus, as $\dot{\sigma}(s) = c_2\sigma(s)$ we can write

$$
\text{SD}(v; z, s) = c_2\mathbb{E}_{p_s^{\text{IMCF}}}[|\sigma(s)\nabla \cdot (\epsilon_s - \epsilon_s^{\text{IMCF}}) + (\epsilon_s^{\text{IMCF}} - \epsilon_s) \cdot \epsilon_s^{\text{IMCF}}|] \tag{C.2}
$$

**For simplicity we compute and report** $\text{SD}(v; z, s)$ **using** $c_2 = 1$. This is such that the "schedule deviation" at a given noise level $\sigma(s)$ can be computed independent of the upper and lower bounds on $\sigma$.

**Empirical Estimation of Optimal Transport Distances:** We use $N = 128$ samples from each sampler, for each conditioning value, to estimate the 1-Wasserstein (i.e. earth-mover distance). Computations were performed using the Python Optimal Transport toolbox. We used the exact LP-based solution, as opposed to e.g. entropic Optimal Transport using Sinkhorn.

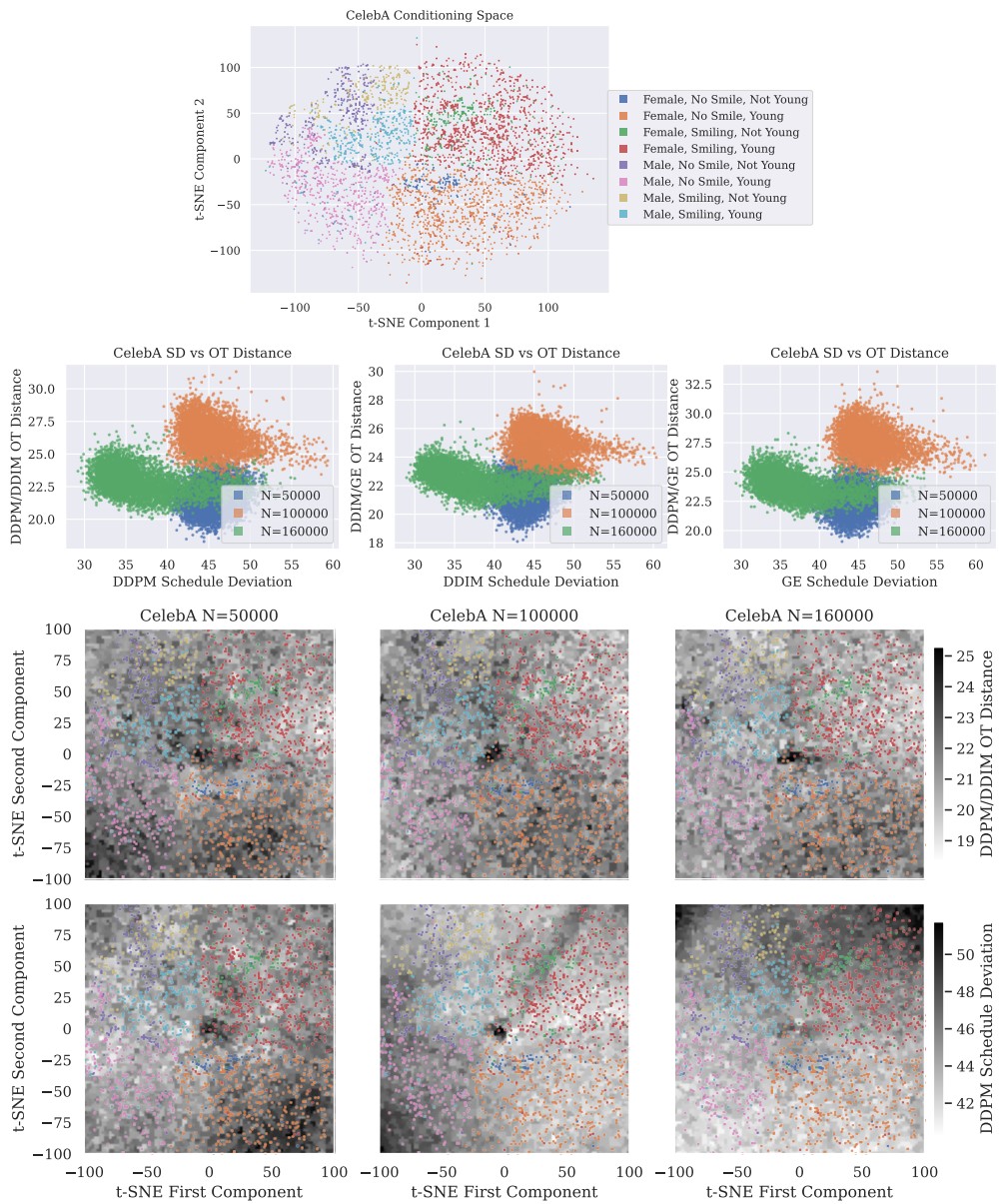

Figure 8: Similar to the MNIST/Fashion-MNIST datasets, we visualize the Celeb-A conditioning space (top) and show the clustering of 3 different attributes (Male/Female, Smiling/Not Smiling, Young/Not Young). Although there is some similarity visually between the OT distance and DDIM/DDPM OT distance, the correlation between Schedule Deviation and OT is weak. We discuss these experiments further in Appendix C.1.

## C.3 Closed-Form Interpolants

Note that we can write the flows $v^\star(x, z = 0)$ and $v^\star(x, z = 1)$ under Eq. (C.1) as

$$v^\star(x, z = 0) = -\frac{\sigma(s)(x+1)}{0.1^2 + \sigma^2(s)}$$

$$v^\star(x, z = 1) = -\frac{e^{-\frac{-(x-1)^2}{2(0.1^2+\sigma^2(s))}}}{e^{-\frac{(x-1)^2}{2(0.1^2+\sigma^2(s))}} + e^{-\frac{x^2}{2(0.1^2+\sigma^2(s))}}} \left(\frac{\sigma(s)(x-1)}{0.1^2 + \sigma^2(s)}\right)$$

$$- \frac{e^{-\frac{-x^2}{2(0.1^2+\sigma^2(s))}}}{e^{-\frac{(x-1)^2}{2(0.1^2+\sigma^2(s))}} + e^{-\frac{x^2}{2(0.1^2+\sigma^2(s))}}} \left(\frac{\sigma(s)x}{0.1^2 + \sigma^2(s)}\right)$$

**Discrete-Support Interpolant.** For the discrete support dataset (Figure 7, upper), we use the interpolant based on Theorem 2:

$$v(x, z) = (1 - z) \cdot v^\star(x, z = 0) + z \cdot v^\star(x, z = 1)$$

**Continuous-Support Interpolant.** For the distribution with continuous support (Figure 7, lower), we take inspiration from Theorem 3, which suggests that for uniform densities, the flow $v(x, z)$ should be convolved with the kernel

$$\int_\xi \frac{e^{2\pi i \xi z}}{1 + \lambda \|\xi\|^4} \mathrm{d}\xi. \tag{C.3}$$

We use the approximation $\mathcal{F}^{-1}\left[\frac{e^{2\pi i \xi z}}{1+\lambda\|\xi\|^4}\right] \approx \frac{c_1}{(1+c_2 z^2)^{3/2}}$. We note this has the same tail behavior, as the Fourier transform of $\frac{1}{\|\xi\|^4}$ attenuates with $\frac{1}{z^3}$. In particular, we use $c_1 = 1.5, c_2 = 16$ for the associated experiments. Thus, we use,

$$v(x, z) = \frac{\gamma(x)}{\gamma(x) + \gamma(1-x)} v^\star(x, z = 0) + \frac{\gamma(1-x)}{\gamma(x) + \gamma(1-x)} v^\star(x, z = 1)$$

where $\gamma(x) = 1.5(1 + 16x^2)^{-3/2}$.

## C.4 Datasets

We visualize the conditioning spaces for the datasets in Section 3 in Figure 1 and show the conditioning space for the CelebA dataset in Figure 8.

**MNIST/Fashion-MNIST**: For the MNIST and Fashion-MNIST datasets, we t-SNE [Van der Maaten and Hinton, 2008] the images to construct the two-dimensional latent spaces seen in Figure 1. We used SciKit-Learn implementation (which uses a Barnes-Hut style approximation for large datasets) with a perplexity of 30 and an early exaggeration of 12 for both MNIST and Fashion-MNIST.

**CelebA**: We use a setup similar to the MNIST/Fashion-MNIST for the CelebA dataset, except that we (a) downsample the images to 64x64 and (b) t-SNE the 40 *discrete attributes* provided by CelebA (using a hypercube), as opposed to the images themselves. We visualize the resulting embedding in Figure 8.

**Maze Solutions Dataset:** The maze dataset (Figure 1, left) consists of maze solutions from different starting points to the center of the maze. We use the fixed maze depicted in Figure 1. The row of the starting cell is picked uniformly at random and the column $c$ (indexed at 0) is picked with probability $p(c) \propto e^{-c/2} + e^{-(7-c)/2}$, i.e. we are more likely to pick starting points near the end in order to start further from the goal point. For each starting point, we randomly sample a solution $s$ based on its length compared to the optimal solution such that $p(s) \propto e^{-(\ell(s)-\ell(s^\star))}$, where $\ell(s)$ is the length of $s$ and $\ell(s^\star)$ is the length of the shortest path to the origin. For a given solution, we then construct a Bezier curve which fits control points along the trajectory which have been slightly perturbed by noise $w \sim \mathcal{N}(0; 0.04)$. We then take 64 evenly spaced points along the Bezier curve use this as the final "trajectory" in the maze which we attempt to generate.

### C.5 Model Architectures and Hyperparameters

For all experiments we used the $\epsilon$-parameterization introduced in Ho et al. [2020] and a "variance-exploding" setup for the Diffusion Schedule as detailed in Appendix B. In particular, we use a log-linear noise schedule where $\sigma(s) = c_1 e^{c_2 s}$, with 512 training timesteps (and 64 sampling timesteps) ranging from $\sigma = 5 \times 10^{-4}$ to $\sigma = 5$ for the experiments in Section 3 and $\sigma = 0.01$ to $\sigma = 35$ for the CelebA experiments.

For the toy dataset in Section 4, we used 1024 training timesteps (and 128 sampling timesteps) with noise values ranging from $\sigma = 8 \times 10^{-3}$ to $\sigma = 10$.

**MNIST/Fashion-MNIST:** We use a U-Net with 5.9 million parameters as the "default" for the MNIST and Fashion-MNIST experiments. We use the same UNet architecture described in Dhariwal and Nichol [2021] with a base channel dimension of 64 and using 2 ResNet blocks per downsampling/upsampling step. We used GroupNorm, with 32 channels per group, for the ResNet normalization layers.

After the first downsampling step we use 128 dimensions (2x the "base channels"). We additionally include an attention block before the 3rd downsampling layer.

Conditioning on the time $s$ and conditioning value $z$ is performed via first embedding each into a 256 dimension (4x "the base channels") into an MLP and a 2 layer MLP. The time $s$ is fed in using a sin/cos embedding.

In Figure 5, we show an ablation where we increase the base channels to $[96, 128, 160]$, corresponding to 13.3M, 23.5M, and 36.8M parameters respectively.

For both MNIST/Fashion-MNIST we train the model using AdamW (with weight decay $1 \times 10^{-4}$) and a cosine decay schedule with an initial learning rate of $3 \times 10^{-4}$ over $300,000$ total training iterations and a batch size of 256 samples.

**Maze Solutions:** For the Maze solutions, we consider a similarly constructed UNet to the MNIST/Fashion-MNIST, but using 1-D convolutions instead of 2-D convolutions. Our training parameters are also similar to the MNIST experiments, but we use instead $100,000$ iterations, an initial learning rate of $5 \times 10^{-4}$, and a batch size of 128.

**Toy Data:** For the toy datasets in Figure 6, we consider a 5 layer MLP with a hidden dimension of 64, input dimension of 2 (value + conditioning) and output dimension of 1 ("denoised value"). The time value is first embedded using sin/cosine and then mapped to a 64 dimensional vector. For each layer in the MLP, we modulate the activations using a FiLM [Perez et al., 2018] conditioning scheme.

For training we use AdamW with $10,000$ iterations, cosine decay with an initial learning rate of $4 \times 10^{-3}$ and a weight decay of 0.01. We use a batch size of 128 and generated synthetic datasets of size $N = 100,000$ samples as described in Section 4 and shown in Figure 7.

### C.6 Full Main-body Experiment Set with Additional Samplers

For completeness, we include additional visualization for the DDPM Ho et al. [2020], DDIM Song et al. [2020a], and Gradient Estimator (GE) Permenter and Yuan [2023] sampling algorithms (described in Appendix B) for each of the MNIST (Fig. 12), Fashion-MNIST (Fig. 13), and Maze solution (Fig. 14) datasets.

Additionally in Figure 10 we show scatter plots analogous to those in Figure 2 for all sampler/dataset combinations we evaluate. In Figure 11 we show an ablation over training samples and per-class Schedule Deviation distributions for the Fashion-MNIST dataset.

## D Broader Impacts and Limitations

**Broader Impact:** We believe that Schedule Deviation is an important step towards understanding the generalization behavior of conditional diffusion models. This may yield insights into downstream phenomena such as hallucination and the synthesis of completely "novel" samples. The insight that conditional diffusion models generally do not denoise also has implications for the design of future sampling algorithms, and more broadly cautions against making strong assumptions on the properties

of learned models in generative settings, irrespective of the original objective inherent in the training loss (in this case, for the model to "denoise").

We hope that these results will motivate greater theoretical and empirical study of non-denwoising flows and, specifically, phenomena such as self-guidance. The exact effect of classifier-free guidance remains poorly understood, despite widespread adoption and deployment. This work highlights that better theoretical and practical understanding of different flow composition rules can potentially yield insights into the behavior of trained models.

**Limitations:** Our proposed metric, Schedule Deviation, has a number of limitations. Namely, it requires sampling from $p_0(x|z)$ (i.e. running the reverse process of a chosen sampling algorithm) and necessitates estimating both (1) the divergence of the neural network and (2) the gradient of the $p_0(x|z)$ noise distribution. Estimating the gradient for small noise levels can require a large number of samples to do accurately, as the variance of the $\nabla \log p_s(x|z)$ estimator increases as $s \to 0$. Furthermore, computing the divergence through the neural network using back propagation is as expensive in practice as computing the full $n \times n$ input-output Jacobian, as it requires $n$ Jacobian-vector-product queries to compute. Both of these computational bottlenecks suggest that computing our metric may be difficult for $\mathcal{X}$ of very high dimensions. However, with greater computational resources and alternative methods for computing the divergence, these concerns may ultimately prove negligible.

We demonstrate feasbility on a toy Conditional-MNIST ($d = 784$) problem, but note that even in this setting, computing the Schedule Deviation for the several thousand points needed to create the heatmaps seen in Figure 12 took multiple hours per checkpoint, longer than the time to train the model itself.

Lastly we note that in this work we consider conditional diffusion in three particular contexts: a low dimensional "toy" environment, a maze solving dataset, and a conditional image generation dataset. We hope that the trends we have identified hold in other domains (e.g. audio or video synthesis) but have not thoroughly investigated precisely how universal the Schedule Deviation is in other settingngs. The evidence we present does suggest that the behavior of diffusion models should not be taken for granted.

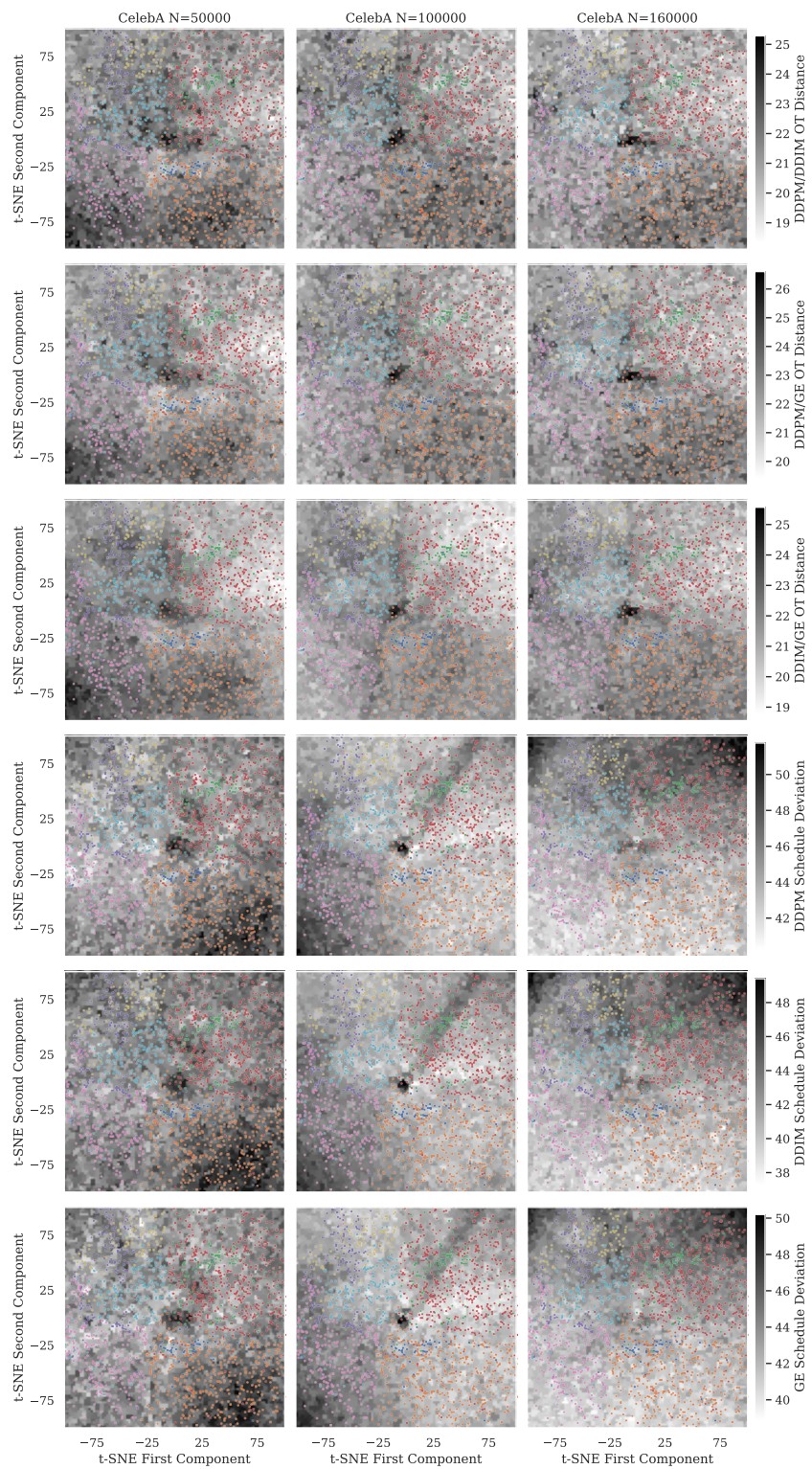

Figure 9: Analogous to Figure 12, we visualize the Schedule Deviation and OT distances for different choices of sampling algorithms.

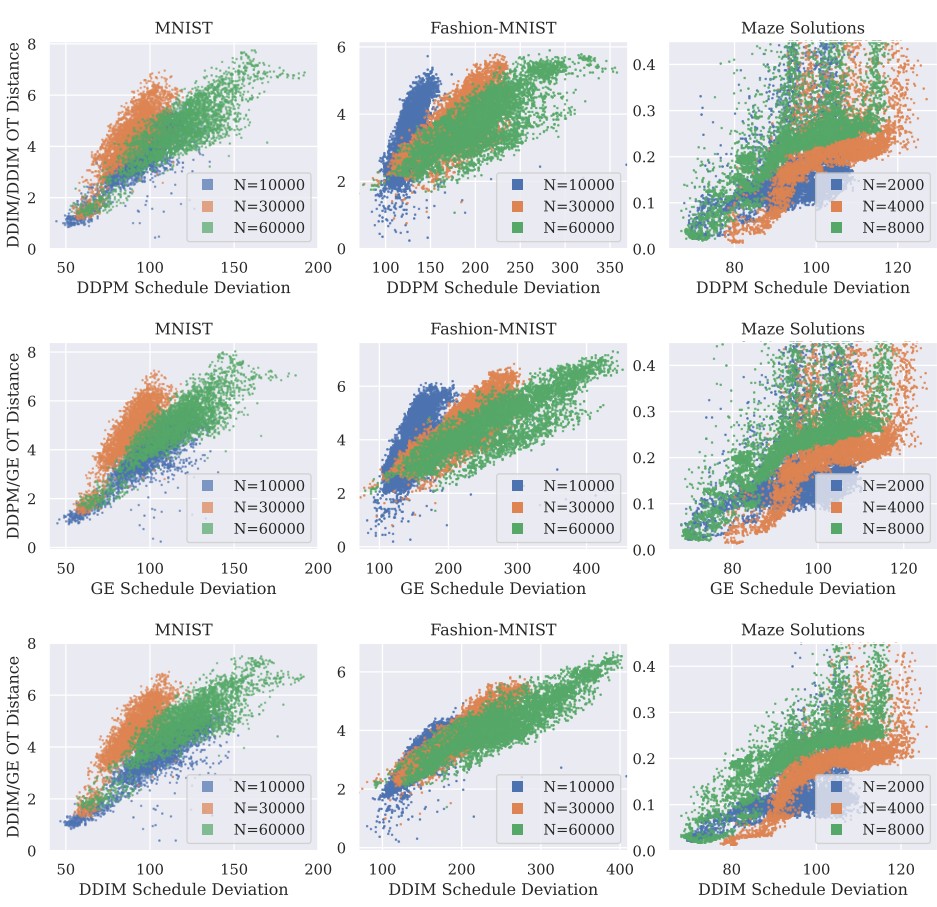

Figure 10: Optimal transport distances vs Schedule Deviation using $p_0$ corresponding to the DDPM, DDIM, and GE samplers, for each of the MNIST, Fashion-MNIST, and Maze datasets.

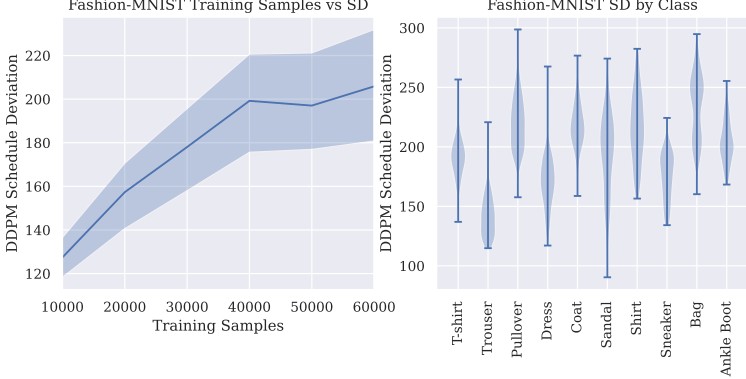

Figure 11: Ablation over training samples and per-class Schedule Deviation for Fashion-MNIST. For the left, 30th, median, and 70th percentiles are visualized for $z$ sampled uniformly over $\mathcal{Z}$.

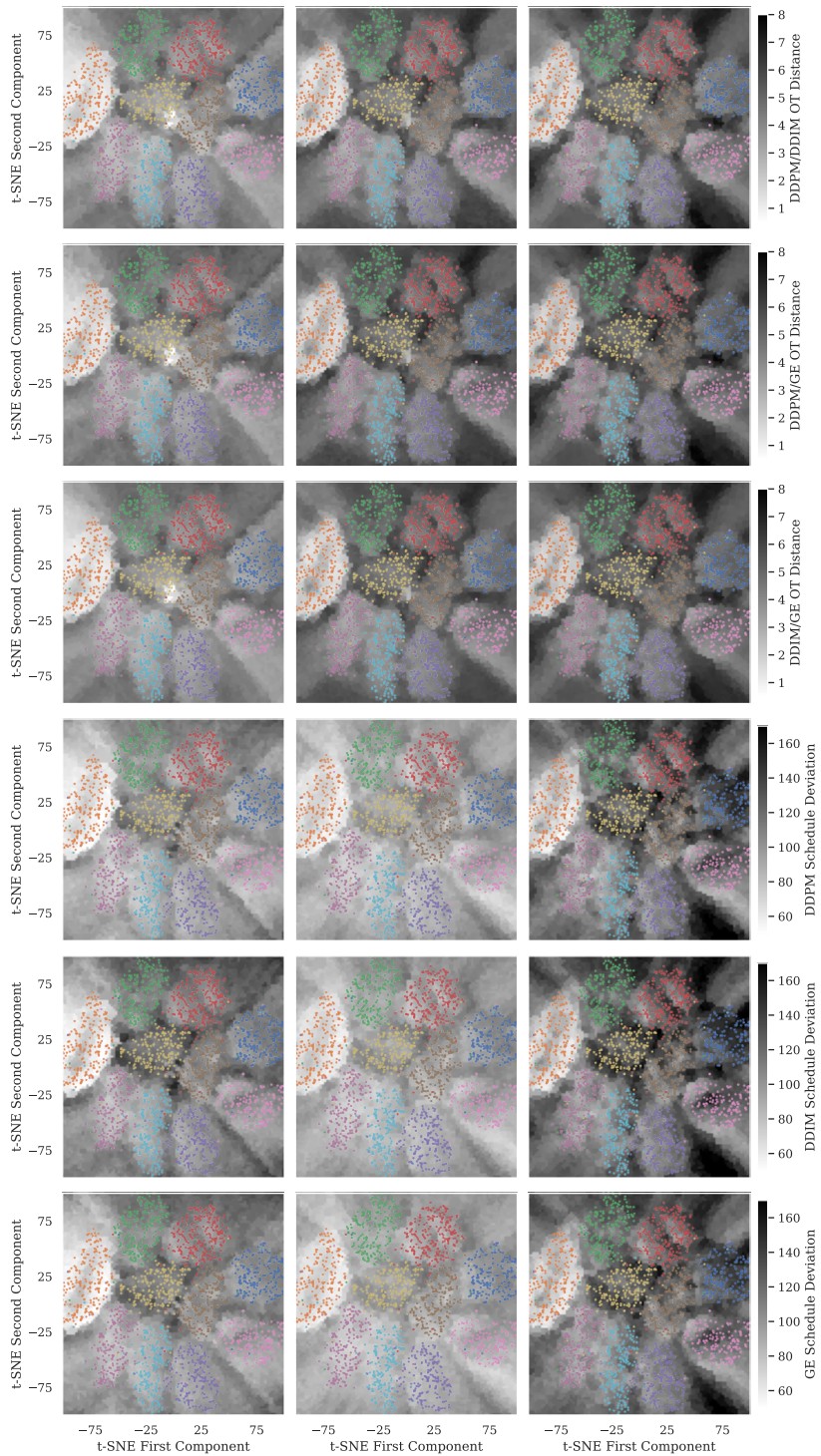

Figure 12: Optimal transport distances (as measured by the empirical 1-Wasserstein distance) and Schedule Deviation for each of the DDPM/DDIM/GE sampling algorithms on the Conditional MNIST dataset.

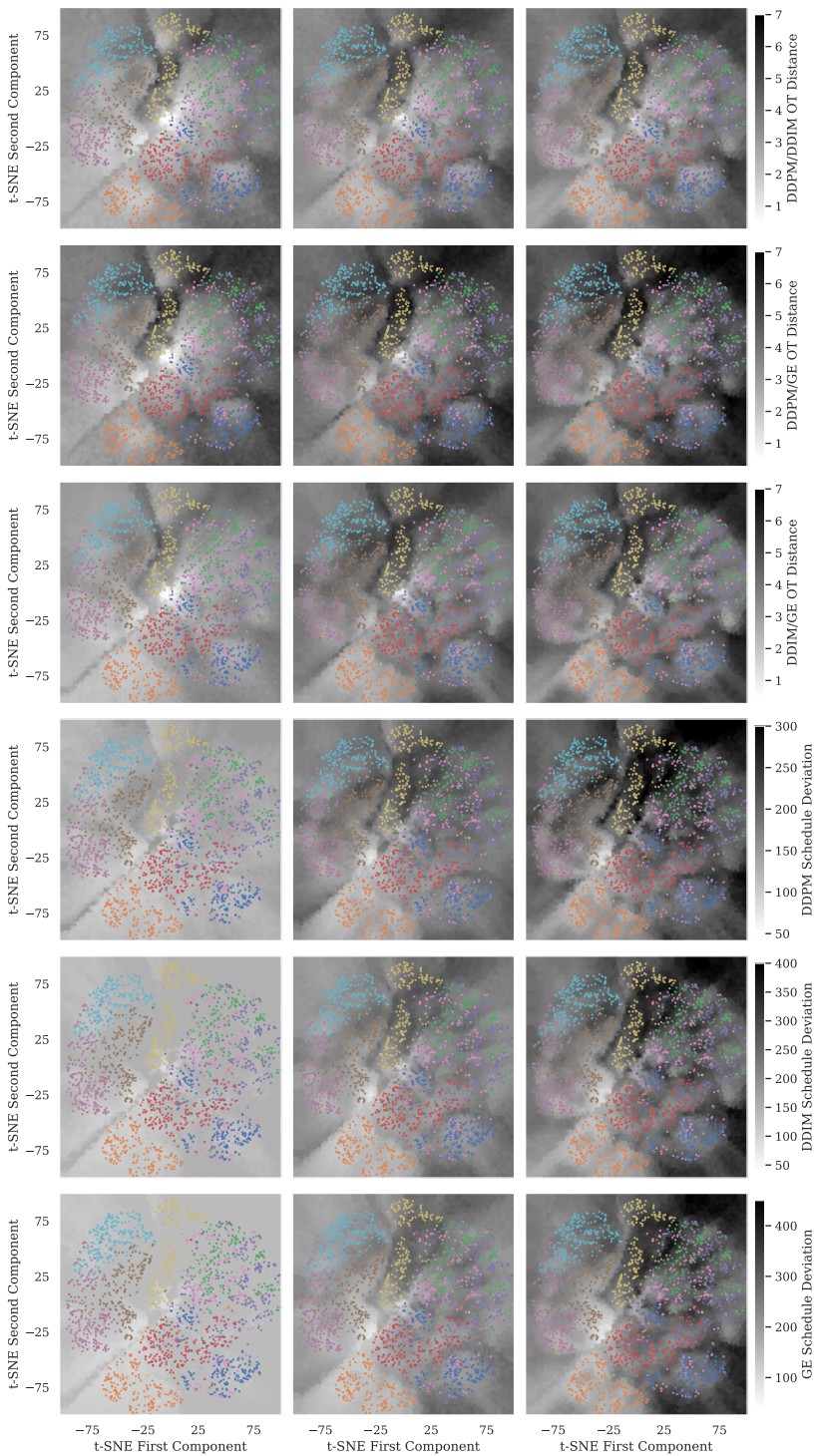

Figure 13: Optimal transport distances (as measured by the empirical 1-Wasserstein distance) and Schedule Deviation for each of the DDPM/DDIM/GE sampling algorithms on the Conditional Fashion-MNIST dataset. Note the per-row scaling on the right differs between sampling algorithms.

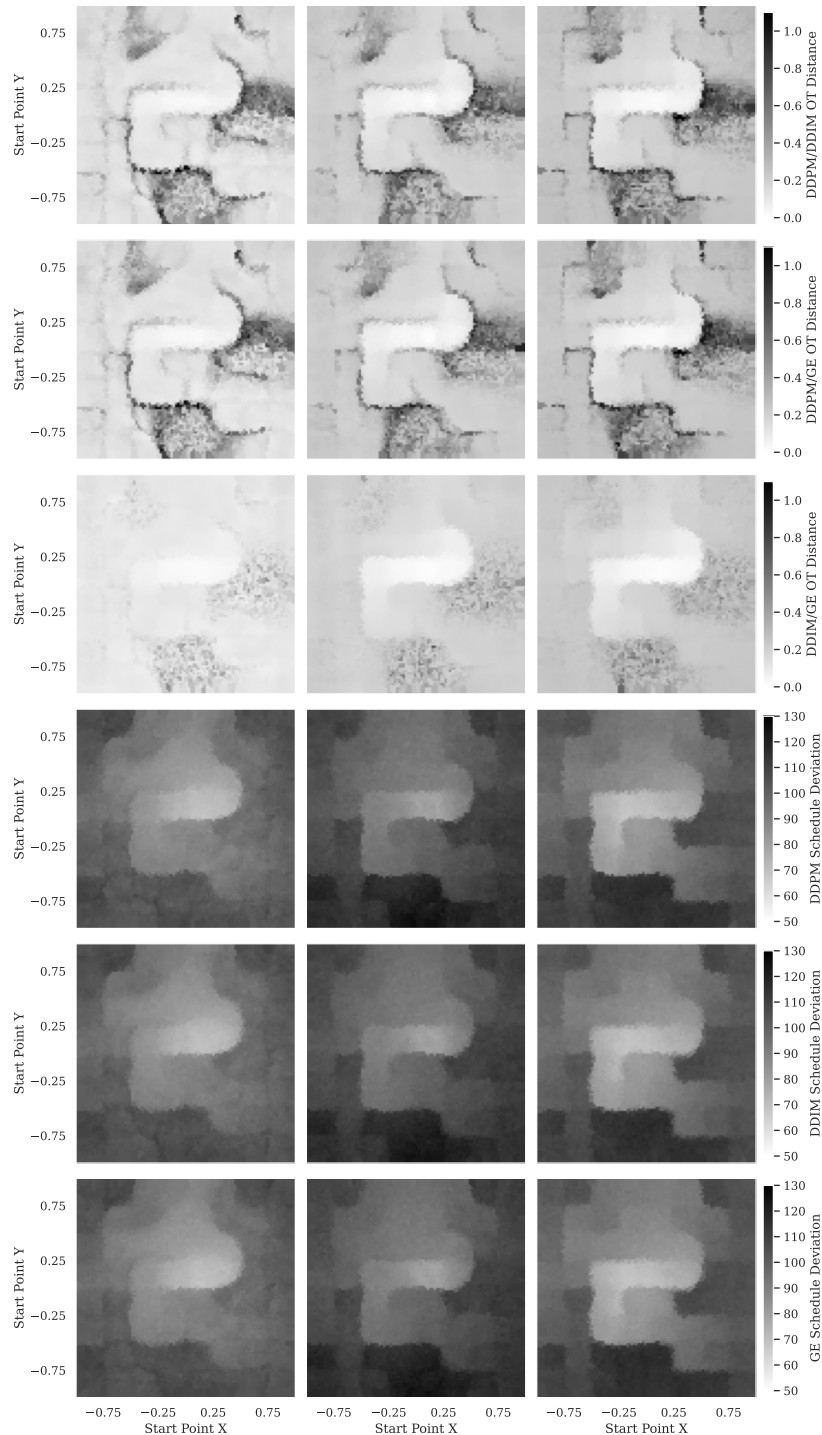

Figure 14: Optimal transport distances (as measured by the empirical 1-Wasserstein distance) and Schedule Deviation for each of the DDPM/DDIM/GE sampling algorithms on the Maze Solutions dataset.

