# OpenReview forum: "Is Your Diffusion Model Actually Denoising?"
_NeurIPS.cc/2025/Conference — NeurIPS 2025 poster_

### Official Review · Reviewer_i2tR · 2025-06-26

**Clarity:** 3
**Significance:** 3
**Originality:** 3
**Rating:** 5
**Confidence:** 3

**Summary:**

The paper proposes a new metric to analyse the sampling behavior of diffusion models, Schedule Deviation. The authors show that the intermediate denoising predictions of trained diffusion models deviate from the predictions of an ideal denoiser, with this effect being consistent across model and dataset scales. To explain it, they posit that neural network smoothness with respect to the conditioning variables leads to unwanted 'self-guidance' in the model, which incorrectly interpolates predicted flows from different conditions.

**Questions:**

- **Unconditional diffusion**: A question that quickly comes up when reading the paper is whether Schedule Deviation exists in unconditional diffusion models, and if so, could it be because of another reason other than the self-guidance due to conditioning presented in the paper?

- **Computing the proposed Schedule Deviation metric**: The authors acknowledge that computing the proposed metric is very expensive. Even for the toy datasets they demonstrate their results on, computing the divergence term $\nabla (v_s - v_s^{\text{IDDF}})$ is too computationally heavy, so they approximate it with a single value from the Jacobian diagonal. This raises the question of whether there's even a chance of demonstrating the paper's findings for practical diffusion models (e.g. text-to-image models).

- **More complex conditions**: Do the same findings regarding self-guidance hold when the ideal denoiser is non-linearly interpolating the conditions? As the conditioning increases in dimensions ($2$-dim in the paper experiments vs $77\times1024$-dim for text) and becomes more complex, it is unlikely that the model is simply linearly interpolating conditions to generalize to unseen prompts. Would you get similar results when interpolating across some learned geodesics on the condition manifold?

**Ethical Concerns:**

["NO or VERY MINOR ethics concerns only"]

**Final Justification:**

The paper presents an interesting new metric to measure the deviation of the trained and an ideal denoiser, which is shown to correlate with observed deviations (DDPM vs DDIM samples) in the model. They also come up with a theoretical motivation for this discrepancy, based on the smoothness of the networks used to implement the denoiser. While the scale of the experiments is limited, this work could be a meaningful stepping stone towards better understanding the inner workings of diffusion models.

**Limitations:**

Yes, limitations have been addressed.

**Quality:**

3

**Strengths And Weaknesses:**

**Strengths**:
- **Novelty**: The authors devise a novel metric to measure the discrepancy between the predictions of a trained and an ideal denoiser. The proposed metric, Schedule Deviation, is well-motivated and a similar approach has not been discussed in the literature before.

- **Findings**: Using the proposed metric, the authors demonstrate a variety of findings that showcase how trained diffusion models do not perform as expected when drawing samples. Going a step further, the authors provide a possible explanation for this discrepancy, which lies in the smoothness of the conditioning used in these models. These findings persist across datasets/model sizes and sampling algorithms and can be of great importance to the development of future diffusion sampling and training algorithms.

**Weaknesses**:
- **Self-guidance experiments**: The authors posit that the issue of *self-guidance*, or the denoiser linearly interpolating flows from different conditions, leads to a larger Schedule Deviation. Theorems 2 and 3 motivate this assumption by showing that a smoothed denoiser interpolates between discrete and continuous conditions. To translate these findings to trained diffusion models, the authors perform the toy experiments in Figures 5 and 6. There, they show that the neural network and closed-form denoisers draw similar samples, but the similarity in Schedule Deviation is not as evident. Additionally, it is unclear why showing that having similar samples and Schedule Deviation means that self-guidance is true for the trained diffusion models.

- **Scale**: The scale of experiments is limited to toy datasets, which is an inherent weakness of the proposed metric. Computing the Schedule Deviation requires (i) approximating the ideal denoiser with a large enough set of samples drawn from the model and (ii) computing the divergence of the difference between the trained and ideal denoiser scores. Although the arguments made in the paper are mostly sound for the toy experiments shown, there may be interesting findings that deviate from the ones shown in the paper in bigger models, such as text-to-image models. It is unclear whether the proposed metric can be used in larger models.

---

> ### Author Rebuttal · Authors · 2025-07-31
>
> We thank reviewer i2tR for their thoughtful feedback and address their concerns, approximately in order:
>
> 1. **Self-guidance experiments:**
>
>     We acknowledge that we do not directly demonstrate self-guidance in learned models, but instead show the capability of self-guidance to reasonably predict the samples and Schedule Deviation of a learned model.
>
>
>     It is unclear how to demonstrate self-guidance in itself besides comparing generated samples. Using the error in the flow between the closed-form and the learned model is difficult to interpret as there is no “baseline” flow that we can contrast self-guidance to in order to show that the computed error is reasonably low. We agree that showing that self-guidance is inherently “true” of models would greatly strengthen this paper and are interested in any ideas the reviewer has for how to effectively design such an experiment.
>
>     We naturally do not expect self-guidance to be exactly true for larger models, which necessarily must have more complicated inductive biases than purely linear interpolation of different scores–there is likely significant coupling between x and z which, as we note in Remark 4.1, our simple model cannot capture. However, we believe the “self-guidance” model gives useful intuition for why Schedule Deviation may be fundamental to conditional diffusion.
>
> 2. **Scaling/computing the proposed schedule deviation metric:**
>
>    While we acknowledge that our metric is expensive, we emphasize that it is not prohibitively so: we demonstrate scalability up to ~40 million parameter models without issue.
>
>
>    We believe that with sufficient compute resources (we would estimate at most several dozen GPUs and a couple weeks’ time) the metric could provide meaningful evaluation of smaller-scale text-conditioned models (~1 billion parameters for e.g. Stable Diffusion 3).
>
>     Unfortunately, this is beyond the limitations of the resources we can acquire, although we believe it is feasible to scale: the metric is highly parallelizable and has approximately the same memory complexity as performing a single forward-backward pass.
>
>     We use t-SNE-conditioned image generation in this work as a small-scale proxy for larger text-conditioned diffusion models and, following the suggestions of Reviewer zm5H, will incorporate a human-faces dataset conditioned on continuous attributes such as age for our next revision as a better proxy for the continuously-conditioned diffusion task.
>
>     We welcome any suggestions from the reviewer for additional conditional generation tasks they believe would be interesting in the ~100 m parameter regime which we could include in the final revision.
>
> 3. **Unconditional diffusion:**
>
>    Unconditional generation should behave analogously to conditional generation in a region with a high density of conditioning data–that is to say, low schedule deviation. While there is still some Schedule Deviation in these regions (presumably due to smoothing/inductive biases with respect to the x and t inputs that our analysis does not capture), this effect is fairly minimal compared to regions with low-density of conditioning support.
>
>    Because the forward process has support everywhere, with a sufficiently large model and enough training time it will always be possible to exactly model the denoiser of the training dataset and overfit to the training data, e.g. achieve low Schedule Deviation for the unconditional setting. However, for conditional models, we show that Schedule Deviation cannot be easily ameliorated by increasing the training data or the model capacity.
>
>    That is not to say there is no deviation from the IDDF in unconditional settings, just that the flow is believed to much more closely approximate a denoising process. The effect of this kind of smoothing has been studied in prior work such as [Aithal 2024], [Scarvelis 2023]. We believe follow-up work explicitly contrasting Schedule Deviation in conditional and unconditional settings to be of interest.
>
> 4. **More complex conditions:**
>
>    Naturally, we do not expect that larger trained models will interpolate via linear combination of the scores–their true behavior is likely a much more complex, architecture-dependent non-parametric. We use Section 4 to highlight how natural and simple inductive biases can be fundamentally at odds with denoising in a manner that is distinct from the smoothing effects studied in unconditional models.
>
>     Although we present no evidence in this paper that self-guidance is directly predictive for larger models, the widespread use and strong empirical performance of classifier-free guidance suggests that self-guidance is an excellent inductive bias. We suggest that better theoretical understanding of classifier-free guidance may be applicable to conditional models as well.
>
>     For higher dimensions, we believe the correct primitive to consider is some complicated nonlinear spline rather than a learned geodesic, as we cannot expect the flow along a path to be purely a combination of its two endpoints. We are considering follow-up directions for larger models motivated by the concept of self-guidance along the lines of what you suggest: namely, can we express a conditional score in terms of a finite basis of unconditional functions, with learned interpolating coefficients? Whether conditional diffusion models interpolate their flows in some low-dimensional manifold is an interesting hypothesis. We hope to study this question in future work.
>
> [Aithal 2024] Aithal, S. K., Maini, P., Lipton, Z., & Kolter, J. Z. (2024). Understanding hallucinations in diffusion models through mode interpolation. Advances in Neural Information Processing Systems, 37, 134614-134644.
>
> [Scarvelis 2023] Scarvelis, C., Borde, H. S. D. O., & Solomon, J. (2023). Closed-form diffusion models. arXiv preprint arXiv:2310.12395.

---

> > ### Comment · Reviewer_i2tR · 2025-08-03
> >
> > Thank you for your detailed responses. Scaling up to compute the proposed schedule deviation metric on very large models (SD-scale) still seems computationally prohibitive, but this does not diminish the significance of your results in more controlled settings. A CelebA-64 experiment, with individual facial attributes as conditions, could greatly help the presentation of your paper.
> >
> > Since most of my questions/concerns have been adequately addressed in the responses you have provided, I will be increasing my score to reflect that.

---

### Official Review · Reviewer_jj8Z · 2025-07-03

**Clarity:** 2
**Significance:** 2
**Originality:** 3
**Rating:** 5
**Confidence:** 3

**Summary:**

This paper investigates a fundamental mismatch between the theoretical formulation of conditional diffusion models as denoisers and their practical behavior during inference. It introduces a novel metric called **Schedule Deviation (SD)** to measure how much the learned generative path deviates from the idealized denoising diffusion path (IDDF). The authors find that **conditional diffusion models routinely deviate** from their idealized dynamics—this deviation persists even with more data or larger models and is theorized to result from **inductive biases favoring smoothness** in the conditioning space. The paper provides both empirical and theoretical analyses, showing SD correlates with sample quality degradation and divergence between popular sampling algorithms like DDPM and DDIM.

**Questions:**

- Why introduce a new metric? Why not just try to find a way to efficiently compute TV distance? Would TV distance also have similar explanatory power?
- How difficult would it be to compute TV vs SD ?
- What precedent in the SDE literature exists for a metric like SD? Can you show you have no reinvented the wheel?

**Ethical Concerns:**

["NO or VERY MINOR ethics concerns only"]

**Final Justification:**

The authors have provided adequate explanation of the significance, motivation, and intuition behind their results. Response to other reviewer's concerns about experiments were also satisfactory. Thus this reviewer recommends an accept.

**Limitations:**

Yes

**Quality:**

3

**Strengths And Weaknesses:**

Strengths:
- High relevance: addresses a core assumption in conditional diffusion
- Originality: novel explanation of the workings of diffusion models
- Rigorously defines a principled and tractable metric (Schedule Deviation)

Weaknesses:
- Lack of motivation and intuition for new metric, Schedule Deviation (SD)
- No survey or connection to literature on SDEs

---

> ### Author Rebuttal · Authors · 2025-07-31
>
> We thank the reviewer for their review and questions. We address your concerns in order:
>
> 1. **Motivation:**
>
>     We consolidate the different motivations and interpretations of our metric below and will be sure to revise our introduction and preliminaries to make the motivation more explicit:
>
>     **We address shortcoming with metrics developed for unconditional settings:** It is believed the score-smoothing/non-denoising behavior of models is fundamentally related to generalization in unconditional diffusion [see Scarvelis, Aithal 2024]. However, existing metrics such as the “error from the ideal closed-form denoiser” ($||v^\star_t(x) - v_t(x)||$) [see Scarvelis] require samples from the true data distribution $p^\star(x)$ to compute $v^\star_t(x)$. For conditional diffusion (particularly with continuous conditioning values) we may potentially have either one or zero samples for a given “z” value and therefore cannot estimate the ideal denoiser for $p^\star(x|z)$. This is what principally motivates the construction of a new metric which can be evaluated even for conditional models.
>
>     Existing metrics are ad-hoc designed in other ways. Namely, while the true denoiser/flow $v^\star$ can be estimated from the training data, there are many choices for the distribution over $x$ on which to evaluate $||v^\star_t(x) - v_t(x)||$. It is unclear whether to use (1) reverse process under the learned v_t starting from noise or (2) the forward process using samples from $p^\star(x)$.
>
>     We argue that the **principled** choice of distribution over x to consider is the one induced by (1) running the full reverse process under the learned $v$ and then (2) subsequently re-noising the generated samples using the forward process. We use $v^{IDDF}$, “the idealized flow” to refer to the **flow for the forward process applied to the p(x | z) distribution of the reverse process under the learned v**. Crucially, we can compute $v^{IDDF}$ without access to $p^\star(x|z)$ using only the learned $v$, meaning our metric can be computed for conditional models.
>
> 2. **Intuition:**
>
>     Note that while we evaluate the Schedule Deviation using Proposition 3.1 (which resembles earlier metrics), we cleanly derive the form in Proposition 3.1 from the instantaneous rate of deviation of $p^v$ from $p^{IDDF}$ as defined in Definition 3.1. In other words, for some intermediate timestep $t$, if we start at $p^{IDDF}_t(x)$, the Schedule Deviation measures the total rate at which the density under $v^{IDDF}$ diverges from the same density evolved under $v$. We believe this is much easier and more intuitive than the “error from a reference denoiser” as in prior work.
>
>     Proposition 3.1 gives an alternate interpretation of our metric as the error between $v$ and $v^{IDDF}$ along the direction of $\nabla \log p^{IDDF}_t(x)$, plus a divergence-based correction term. This reveals that not all forms of error between $v$ and $v^{IDDF}$ cause deviation from a denoising probability path. At first order, it is only the error along $\nabla \log p^{IDDF}_t(x)$ which causes deviation, a property which our metric makes immediately clear. This makes intuitive sense, as the transport equation (Eq A.1) implies that adding a “curl” component to $p_t(x)v_t(x)$ has no effect on the evolution of the density $p_t(x)$, i.e. $v$ can be arbitrarily far from $v^{IDDF}$ without affecting the density.
>
>    Lastly, as we note just before Theorem 1, our metric resembles the advantage function used in the storied Performance Difference Lemma from reinforcement learning, applied to the evolution of densities. Indeed, an extension of the Performance Difference Lemma is central to the proof of Theorem 1 found in Appendix A.
>
>  3. **Novelty and Connection to SDE Literature:** To our knowledge these contributions above are all novel and we are unaware of a similar metric in either the SDE, optimal transport, or diffusion literature beyond the aforementioned prior works.
>
>     We base our formulation of diffusion on the Stochastic Interpolants and flow-matching frameworks [Albergo, Lipman], introducing diffusion models using its flow-based ODE formulation rather than the (equivalent) SDE formulation as it simplifies the notion of having a non-denoising flow and are generally easier to reason about the SDEs, which require significantly more technical mathematical treatment.
>
>     Note that we additionally provide SDE-based sampling procedures and equivalent formulations in Appendix B for interested readers who wish to understand SDE-based sampling works in the context of our paper. We will signpost to this appendix for readers who are interested in this.
>
>     We are of course happy to reference the connections to the SDE-based literature beyond the original diffusion works by Sohl-Dickenstein, Ho, and Song, which were of course principally SDE-based. We also will create an extended related works in the appendix with the relevant SDE related work.
>
> 4. **Why not estimate TV directly?**
>     Estimating the TV distance directly requires calculating the difference in densities between $p_t$ and $p_t^{IDDF}$ over the entire space x. As we cannot integrate over the space of all images, it is not easy to compute the TV distance directly. Estimating the associated densities $p_t(x)$, $p_t^{IDDF}$ (or their ratios) is also numerically unstable in practice.
>
>     In contrast, our method for computing Schedule Deviation only requires sampling x from $p_t^{IDDF}$, meaning we can restrict our computation to areas with high likelihood. Furthermore, the quantities involved are all log-densities or their gradients,
>
>     Note that $TV(p_t^{IDDF} | p_t)$ depends only on $v_s(x)$ for $s > t$, not $v_t(x)$ itself. Thus, given only the flow $v_t(x)$ at time $t$, it is more intuitive to relate the value of $v_t(x)$ to the different in the *change* of the density $p_t$ compared to $p_t^{IDDF}$, which is precisely what Schedule Deviation captures.
>
> [Albergo 2023] Albergo, M. S., Boffi, N. M., & Vanden-Eijnden, E. (2023). Stochastic interpolants: A unifying framework for flows and diffusions. arXiv preprint arXiv:2303.08797.
>
> [Aithal 2024] Aithal, S. K., Maini, P., Lipton, Z., & Kolter, J. Z. (2024). Understanding hallucinations in diffusion models through mode interpolation. Advances in Neural Information Processing Systems, 37, 134614-134644.
>
> [Lipman 2022] Lipman, Y., Chen, R. T., Ben-Hamu, H., Nickel, M., & Le, M. (2022). Flow matching for generative modeling. arXiv preprint arXiv:2210.02747.
>
> [Scarvelis 2023] Scarvelis, C., Borde, H. S. D. O., & Solomon, J. (2023). Closed-form diffusion models. arXiv preprint arXiv:2310.12395.

---

> ### Comment · Reviewer_jj8Z · 2025-08-07
>
> The authors have adequately addressed this reviewers concerns about the motivation and intuition behind the authors results.
>
> Furthermore, after consideration of the authors response to the other reviewer's concerns about experiments, the score will be bumped up by 2.

---

### Official Review · Reviewer_zm5H · 2025-07-04

**Clarity:** 2
**Significance:** 2
**Originality:** 3
**Rating:** 4
**Confidence:** 4

**Summary:**

This paper derived a metric Schedule Deviation from theoretical motivations, measuring the difference of the probability path from the ideal or theoretical ones. Then they derived a tractable way to estimate it from expectation. Using this metric they measured the schedule deviation in some practical conditional diffusion models, and found it’s prevalent and related to smoothing w.r.t. conditions. Finally they derived a closed form score smoothened w.r.t. condition, and checked empirically that neural networks exhibit similar generalization behavior and schedule deviation from this closed form.

**Questions:**

### Questions

- One major concern is, The Schedule Deviation is nice and novel, but seems there are some simpler and easier to compute metrics that could do similar things, that authors didn’t mention or compare.
    - For example, one simple ways is to compare the **deviation of neural score function from “ideal” score**. Recently a few papers have computed this through the diffusion trajectory, they found non-negligible deviation esp. at low noise scale and they found this is exactly the reason why diffusion generalize beyond training set. ( for uncond diffusion [Wang, Vastola (2023/2024), Li, Dai, Qu, (2024)]).
    - It seems the identity in Prop 3.1 is different but quite related to averaged score deviations Eq.28 in [Wang, Vastola (2024)], authors may want to discuss this link. The way authors compute it is also quite similar, basically assuming a delta mixture model of the underlying distribution.
    - Per Girsanov theorem, I think score deviation or velocity deviation can also leads to bound in distribution difference.
    - Maybe the authors could plot this score deviation on X axis for Figure 4, I imagine the effect will be similar.
- Not really sure how Schedule Deviation could be computed in Figure 5 for condition values outside the support of training data. When the condition z is out of training set, how do you compute the ground truth score and velocity and compare to neural network? (as described in L183-188)
- Could the Theorem 3 be generalized to more complex topology of Z set? like MNIST case, it’s 2d embedding or some high dim embeddings of classes?
- Did the authors used CFG in their main experiments? Should we describe the effect of it on schedule deviation, since it’s so commonly used in conditional diffusion model?

- Regarding the score smoothing idea, authors may also cite the recent theoretical treatment in [Chen 2025] and empirics in [Wang, Vastola (2024)].
    - More generally I feel smoothing the velocity field will leads to schedule or score deviation, either to x or z.

[Chen 2025] Chen, Z. (2025). On the interpolation effect of score smoothing. *arXiv preprint arXiv:2502.19499*.

[Wang, Vastola (2024)] The Unreasonable Effectiveness of Gaussian Score Approximation for Diffusion Models and its Applications. TMLR *arXiv:2412.09726*.

**Ethical Concerns:**

["NO or VERY MINOR ethics concerns only"]

**Final Justification:**

I think in the most recent round of rebuttal the authors have successfully addressed most of my technical concerns about the paper! As long as the authors improve the clarity of the paper by moving some text & figures from the appendix to the main text in the final version, I feel it's ready for acceptance.

**Limitations:**

yes in appendix D.

**Quality:**

3

**Strengths And Weaknesses:**

### Strength

- Conditional diffusion models and esp. classifier free guidance have been less theoretically examined than unconditional one. So the subject is quite interesting. The conceptual point of conditional diffusion not denoising is interesting. (though in a sense
- The observation of discrepancy between scheduler / solvers and schedule deviation (or score difference in my reading) is interesting and relevant to practical usage.
- Evaluated of a few different settings.
- Smoothness with respect to the conditioning variable is interesting and less studied before. So the theorem 2 is a nice contribution to the field.
- Theorem 3 is nice extension of closed form diffusion (score-smoothing) model to conditional diffusion. It also illustrates these smooth score can be understood as consequence from certain regularization / constraint Eq. 4.1. This is quite aligned with some recent work of how some inductive biases / constraint of diffusion induces the actual neural score and samples. [Kamb, Ganguli, 2024; Niedoba 2024]
- The conceptual link of deviation of path from ideal one and score smoothing w.r.t. conditioning label is interesting (previously more people think about smoothing w.r.t. x).

[Kamb, Ganguli 2024]: Kamb, M., & Ganguli, S. (2024). An analytic theory of creativity in convolutional diffusion models. *arXiv preprint arXiv:2412.20292*.

[Niedoba 2024] Niedoba, M., Zwartsenberg, B., Murphy, K., & Wood, F. (2024). Towards a Mechanistic Explanation of Diffusion Model Generalization. *arXiv preprint arXiv:2411.19339*.

### Weakness

- The conditional flow notation / framework is quite elegant but a bit confusing… the equivalent form showed in Proposition 2.1 is more common and familiar to the diffusion field. Eq. 2.4 and 2.5 are commonly found in Karras 2022 (EDM paper) etc. Maybe authors should cite them there too.
    - For example, the IDDF seems not different from the velocity field determined by the “ideal / true score”. Though what is true score is still tricky, if we have only finite samples from dist.
- Empirically the connection of Schedule deviation and DDPM DDIM distance is clean in the image case, but quite fuzzy in the maze case (Fig2 left and Fig4 right). This made us question the generality of this connection. Maybe further experiments could clarify when this work.
- Theoretically, not sure if the proposed Schedule deviation metric cannot be substitute by some simpler and easier to compute metrics, e.g. difference of neural score from the ideal / true score. (details see questions)

- One conceptual issue for the paper is that IDDF as defined in the paper is based on the empirical score of the training set, and being equal to that will render the model only able to sample from the delta mixture of the training set (as noticed by many authors [Scarvelis 2023], [Wang Vastola 2023], [Li, Dai, Qu, (2024)]). So opposing to L225, deviation from IDDF (as defined in the paper) may be a problem, but being equal to IDDF is not a solution either.
    - The paper above found the score deviation (esp. at low noise scale) and they found this is exactly the reason why diffusion generalize beyond training set. So to say that SD is a bad thing to be alleviated is not totally valid.

- The final experiment toy datasets is a bit too toy or simple. I feel at least a MNIST or human face (e.g. FFHQ) experiment could be done to make it more impactful. As you know face and digit datasets have discrete attributes that could be interpolated.

- [Minor] L274 Not really sure why self guidance is called self guidance… it’s more like score smoothing w.r.t. condition. But personal opinion.

Some typos

L145 *consistency → consistently*

L230 *correlated → correlated with*

L287 *presence → presence of*

[EDM] Karras, T., Aittala, M., Aila, T., & Laine, S. (2022). Elucidating the design space of diffusion-based generative models. *Advances in neural information processing systems*, *35*, 26565-26577.

[Scarvelis 2023] Scarvelis, C., Borde, H. S. D. O., & Solomon, J. (2023). Closed-form diffusion models. *arXiv preprint arXiv:2310.12395*.

[Wang Vastola 2023] Wang, B., & Vastola, J. J. (2023). Diffusion models generate images like painters: an analytical theory of outline first, details later. arXiv preprint arXiv:2303.02490.

[Li, X., Dai, Y., & Qu, Q. (2024)] Understanding generalizability of diffusion models requires rethinking the hidden gaussian structure. NeurIPS

---

> ### Author Rebuttal · Authors · 2025-07-31
>
> Thank you for the thorough reading and the in-depth review. We will clarify several potential sources of confusion that we realize may not have been sufficiently emphasized in the text.
>
> 1. **Clarifying the Construction of the IDDF:**
>
>     The $p_t(x|z)$ (including $p_0(x|z)$) used in the definition of Schedule Deviation are the marginals consistent with the learned flow $v$, not the true flow $v^\star$. That is to say, $p_0$ arises from the reverse process under the learned model and is *not* the distribution of the training set (which we define as $p^\star(x|z)$ at the start of Section 2). This makes $p_0(x|z)$ and, by extension, $v^{IDDF}$ well-defined even when far from the support of the training data. We address this point explicitly in Remark 3.1.
>
>     Given the understandable confusion, we will (1) move Remark 3.1 into the preliminaries and (2) make the preliminaries less dense to read, with the appropriate references to Karras and other similar frameworks, (3) we will use a different symbol besides $v$ in the definition of Schedule Deviation to denote the learned flow and (4) we will also rename IDDF from “Ideal Denoising Diffusion Flow” to “Model Consistent Denoising Flow” (MCDF) in order to make the distinction between the ground-truth flow $v^\star$ and $v^{IDDF}$ immediately clear.
>
>     Hopefully this mitigates your concern that the IDDF is based on the empirical score of the training set. In fact, because we do not rely on our training set at all, the schedule deviation can be computed solely given the pretrained model, an advantage we will highlight in our revised introduction.
>
> 2. **Positioning with Respect to Prior Work:**
>
>     We use a flow-based formulation as it is simpler to reason about what it means to diverge from the ideal flow and mathematically simpler to manipulate than the SDE formulation. We based our formulation off of [Albergo 2023] but naturally the same framework exists in other works as well. We include an SDE formulation as well in Appendix B for completeness and will of course cite [Karras] as well.
>
>     Previous work (e.g., Scarvelis, Chen) uses smoothing of the ideal denoiser to study generalization in unconditional models. In contrast, we argue that for conditional diffusion, "approximately denoising" is less informative due to the much higher  Schedule Deviation, which we explain by our model of self-guidance. We will include [Wang], [Li], and [Chen] in our prior work and explain how our proposed metric and interpretation of conditional diffusion are fundamentally different from the variants they consider.
>
> 3. **Schedule Deviation and “Fuzziness” on the Maze Dataset:**
>
>     The correlation between Schedule Deviation and DDPM/DDIM for the Maze dataset is still quite strong. The “fuzziness” in Figure 4 is due to when the conditioning is exactly on the walls between the cells in the Maze (see Figure 7), which exhibit extremely high OT distance between DDPM/DDIM exactly on the boundary (Figure 2, upper left) but where the Schedule Deviation (Figure 2, lower left) looks much more clean. Removal of these boundary points makes the association much more clear.
>
>     We believe the noise in the optimal transport distance is due to the OT distance we use (1-Wasserstein distance) being very sensitive to changes in the probabilities for modes that are very far apart, such as when evaluating on the “walls” of the maze. Using “clipped” distances in the cost matrix for the OT distances can remove the associated “fuzziness” you observe, but we did not want to create the appearance of hacking the metric to be more favorable and justify why we used a non-standard metric. In short, we believe this is due to the OT calculation rather than our metric, which does not exhibit such outliers.
>
> 4. **Relation to Simpler Metrics:**
>
>     As you point out, error between the learned model and the ideal flow is sufficient to give an upper bound on the deviation of the probability densities. The key strength of our metric is that it also has a lower bound for the deviation of the probabilities.
>
>     Any simpler metric based on error with the ideal flow cannot yield a lower bound. By the transport equation (Eq A.1), any divergence-free component of  $p_t(x|z)v_t(x|z)$ has no effect on the evolution of the marginals of the reverse process. Thus it is possible to add an arbitrary “curl” component to $v^IDDF$ while preserving the same marginals. Thus the “error” between $v$ and $v^IDDF$ can be arbitrarily high despite the fact that both the intermediate marginals and the final distributions are identical. By contrast, our metric directly measures the instantaneous deviation in how the densities are evolved and is agnostic to any such “curl” components in the flow. We will better explain this fact and improve the motivation for our metric in the preliminaries.
>
>     Comparing our metric empirically to the error wrt $v^{IDDF}$ is an excellent suggestion, and we will develop some simple experiments or examples which contrast the two.
>
> 5. **Regarding the low-dimensional experiments:**
>
>    We welcome the reviewer’s suggestion of using a human-faces dataset, and intend to incorporate an experiment on either FFHQ or CelebA using continuous conditioning attributes for a subsequent revision in addition to our existing experiments in Section 3.
>
>    The datasets for the experiments in  Section 4 are designed to mirror the settings of Theorem 2/Theorem 3, where we can analytically compute the smoothest possible interpolant. Unfortunately deriving a closed-form analogous to Theorem 2 or 3 for the conditioning space of more complex data such as MNIST or FFHQ seems generally infeasible. The setting of Theorem 2 is solvable due to specific properties of cubic spline interpolations and Theorem 3 is due to the uniform weighting of the conditioning distribution analyzed.
>
>     Although Theorem 3 could potentially be generalized to non-uniform densities, the final expression would involve nested Fourier and inverse Fourier transforms in the final expression which cannot be further simplified in the general case. We believe that deriving the closed form under other “toy” conditioning distributions or smoothness penalties (yielding correspondingly different spline interpolations in discrete conditioning support analogous to Theorem 2) may be more pedagogically instructive for understanding biases in conditional diffusion. We believe such investigations would perhaps be a good fit for a future, explicitly theoretically-focused work.
>
>    In the case of MNIST or FFHQ, any predictive model would almost certainly need to incorporate inductive biases of the NN architecture as well. Modeling these biases is out of scope for this work and we direct the reader to [Kamb 2024] which investigates locality biases in purely convolutional architectures. We will be sure to expand on this in our prior work.
>
> 6. **Schedule Deviation vs Fidelity:**
>
>     Schedule deviation measures whether a model is “consistent” in representing the denoiser for its own final distribution. It is possible to construct models with low schedule deviation that poorly model the desired conditional distribution or, vice-versa, models with high schedule deviation that model the distribution extremely well. For this reason we note in Remark 3.1 that schedule deviation is orthogonal to fidelity.
>
>     We do not claim that schedule deviation needs to be mitigated. In fact, given how effective classifier-free guidance is, schedule deviation arising from “self-guidance”-style smoothing may be an excellent inductive bias.
>
> Addressing other miscellaneous questions/comments:
>
> 1. As discussed above, our metric is much stronger than the prior variants used in the literature. We will be sure to expand the prior work to better distinguish the advantages of our approach to justify the additional computational complexity of our approach. We encourage the reviewer to read our response to similar questions raised by reviewer daH1.
>
> 2. We did not use classifier-free guidance in these experiments as we were interested in the inductive biases of learning the conditional score by itself.  It is known that classifier free guidance results in non-denoising paths [Bradley 2024] (also potentially relevant to the effect of guidance is Skreta 2024), so it would certainly be interesting to investigate empirically the Schedule Deviation of either classifier-based or classifier-free guidance. We believe there are many settings where evaluating the Schedule Deviation may be of interest and unfortunately are not able to address all of them in this paper, but will do so in a follow-up work.
>
> 3. We emphasize the term “self guidance” to highlight that better understanding of guidance (which remains poorly understood) may simultaneously yield better understanding of how conditional diffusion models generalize. While it is perhaps intuitive for Theorem 2/3 that smoothness regularization yields a linear combination of the flows at the original datapoints, that this combination is independent of the choice of z is surprising.
>
> [Albergo 2024] Albergo, M. S., Boffi, N. M., & Vanden-Eijnden, E. (2023). Stochastic interpolants: A unifying framework for flows and diffusions. arXiv preprint arXiv:2303.08797.
>
>
> [Kamb 2024] Kamb, M., & Ganguli, S. (2024). An analytic theory of creativity in convolutional diffusion models. arXiv preprint arXiv:2412.20292.
>
>
> [Bradley 2024] Bradley, A., & Nakkiran, P. (2024). Classifier-free guidance is a predictor-corrector. arXiv preprint arXiv:2408.09000.
>
> [Skreta 2024] Skreta, M., Atanackovic, L., Bose, A. J., Tong, A., & Neklyudov, K. (2024). The Superposition of Diffusion Models Using the It\^ o Density Estimator. arXiv preprint arXiv:2412.17762.

---

> > ### Comment · Reviewer_zm5H · 2025-08-07
> > **Thanks for response | follow up questions regarding the metric**
> >
> > We applaud the authors for their super detailed explanation regarding our confusions!
> >
> > [1. **Clarifying the Construction of the IDDF**] This definitely clarifies a lot, we thank the authors for their patience. We apologize for not comprehending the Remark 3.1 deeply. This point the IDDF does not depend on the ground truth flow is indeed extremely important and might led to some of our further confusions down the line. This clarify also the name Schedule Difference!
> >
> > [2. **Positioning with Respect to Prior Work**] Thanks for the discussion on the differences to these work which rely on certain “ideal / ground truth score”.
> >
> > [3. SD **“Fuzziness” on the Maze Dataset**] Thanks for the transparency in discussion this point. Without further details, it’s still quite hard to gain intuitions about this maze trajectory task. Potentially some details could be added to the Figure 2 — currently I have no way to see where is the wall and where is the path. Generally some more intuitive explanation / schematics of the datasets could be helpful, esp. these distributions / generative modeling tasks are not super common. We recommend discussing these or polishing these experiment setups in the future version.
> >
> > [4. **Relation to Simpler Metrics**] Thanks for this point, the theoretical explanation makes perfect sense, i.e. the score / flow deviation will take into account the “curl component” while the proposed metric won’t. The point about SD both upper and lower bound the distribution difference is also well taken, it’s indeed stronger. This could be put in the discussion / remarks to motivate or compare the metric to the prior ones. The empirical experiment comparing the two is indeed interesting.
> >
> > However, even though the metric in theory has much difference, the empirical version of the v_s^IDDF (L643) is very much like the delta mixture scores mentioned in many papers including [Wang, Vastola 2023/24]. I imagine the property of it can be quite similar to the score deviation from gmm score.
> >
> > [5. **Regarding the low-dimensional experiments**] Thanks for the technical discussion on the low dimensional experiments. We agree with the authors that the analytical solution for more complex conditioning signal is quite hard and can be nasty. We agree that seeking closed form solution for other simplified conditioning distribution can be pedagogically interesting in future theory focused work.
> >
> > [6. **Schedule Deviation vs Fidelity:**] Point well taken, indeed now I understand SD is more like a self consistency and it is in principle orthogonal to fidelity.
> >
> > -------
> >
> > Follow up question.
> >
> > We applaud the authors for the extensive rebuttal and explanation on the theory side, it makes much more sense now! We also agree upon the observation that schedule deviation correlate with DDIM, DDPM deviation.
> >
> > But still, the actual quantification / implementation of the metric still feels a bit ad hoc and exotic.
> >
> > 1) A more detailed algorithm box in the final version can be very helpful for practical readers. (many details in Appendix C.1 are very well written, but without a brief version of it in the main text, it’s quite hard to infer how you actually estimate this metric from Prop 3.
> >
> > 2) even though I agree on the correlation, it feels there are so many things that can go into the estimation of the SD and the estimation of OT distance…I’m not sure if these two quantities should be correlated in theory….
> >
> > 3) Is it true that, the SD metric do not just depend on the velocity field / neural network, but also the actual scheduler / sampler / ode solver? it SD from different solvers correlated?  If you compute the Schedule difference induced by DDIM instead of DDPM, how does it correlate with the DDIM DDPM difference?

---

> > > ### Author Response · Authors · 2025-08-07
> > >
> > > Thank you for your reponse and taking the time to produce such thoughtful feedback!
> > >
> > > **On your initial comments:**
> > >
> > > On points 1,2,5,6 we are glad to hear that your concerns have been broadly addressed! We will ensure that the underlying points are clearly expressed in the next revision and will especially improve our introduction/prior work.
> > >
> > > 3. We visualize the same maze evaluated in Figure 2 in Figure 7 of Appendix C, as well as visualize multiple "solution" trajectories from the maze data. As we introduce a new dataset, it would make sense to visualize the maze dataset in the main body (either in a wrapfig next to the description of the dataset, or directly next to the maze SD figure). We ran out of space to do so in our original submission and so put this in the Appendix, but will do a better job of introducing the datasets with an additional page provided for a camera ready submission.
> > >
> > > 4. Our metric does deviate empirically from a simple "score error" type metric, suggesting that the "curl" components of the error may be non-negligibly different. We will include additional experiments to this effect. The reformulation of SD in Prop 3.1 suggests (ignoring the divergence term for simplicity) that it is the error along the $\nabla \log p_s^{IDDF}$ direction that is most salient. Along other directions the error is not important, as only the higher-order differences (as measured by the divergence term) matter for SD. It may be useful to drop the divergence term as a "cheaper" (albeit less theoretically sound) proxy for the Schedule Deviation that is much more scalable as it does not require a divergence computation. Whether or not this is the case, this is an interesting ablation of our metric we will perform for the Appendix.
> > >
> > > **Response to new questions:**
> > >
> > >   1. Thank you for this excellent suggestion. We will add an algorithm box for the final submission to make the computation itself easier to follow and such that it is more explicitly clear what is being proposed and how to replicate our metric.
> > >
> > >   2. We prove rigorously at least one aspect of why sampler OT distance/SD are correlated in theory: that Schedule Deviation is directly correlated with the TV distance between the intermediate marginals $p_t$ and those of a reference noising process $p_s^{IDDF}$. We consider OT distance rather than TV in our experiments as we can tractably estimate OT distance given only samples (using the Sinkhorn algorithm), while TV would require access to densities, which we do not have.
> > >
> > > When the marginals $p_s$ arise from a noising process, DDIM/DDPM traverse the same probability path in the continuous time limit (see the derivation of the SDE formulation which we include for completeness in Appendix B). Thus, for lower Schedule Deviation, we can reason that the OT/TV distance between samplers should go to zero (modulo differences due to discretizing SDEs vs ODEs). Thus we believe SD is a reasonable *upper bound* proxy for divergence between samplers in the continuous-time (high number of steps) limit.
> > >
> > > Whether SD in general also *lower bounds* the difference between samplers (as opposed to lower bounding the TV distance to a denoising path, as shown by Theorem 1) is unclear. We believe that this is at least the case locally along the path, i.e. having high schedule deviation at a time $t$ means that for $s \in [t - \epsilon, t + \epsilon]$ the distributions of $p_s$ under DDPM/DDIM diverge when initialized to $p_t$. However, we cannot rule out that DDPM/DDIM later differ in an "opposing" manner that corrects this "error" from time $t$ at some time closer to $0$ such that the final distribution $p_0$ is the same.
> > >
> > > Because of the ability to "correct" the error incurred at a different timestep, it seems impossible to create a metric is both (1) local to a particular time $s > 0$ and (2) that always lower bounds the difference at time $s = 0$. However, we believe that in practice sampling algorithms don't perform these kinds of "serendipitous" corrections, which is why SD is empirically predictive of differences in $p_0$. It may be possible to construct a family of examples on which the Schedule Deviation and evolved densities under different samplers can be computed exactly such that this could be shown, although it is both unclear how to construct such an instance that is non-trivial and whether such an example would be instructive for models on real-world datasets.
> > >
> > >   3. The SD depends on the sampling algorithm as it depends on a choice of $p_0$ from which to construct the IDDF. We visualize the Schedule consistency for different sampling algorithms  and reproduce all main-body figures for different sampling algorithms in Appendix C. In practice we found these are all quite similar and do not depend on which algorithm is used to generate samples from $p_0$. We can add an additional figure plotting the Schedule Deviations using different samplers against each other to demonstrate this explicitly.

---

### Official Review · Reviewer_daH1 · 2025-07-04

**Clarity:** 1
**Significance:** 3
**Originality:** 3
**Rating:** 5
**Confidence:** 3

**Summary:**

The authors question the denoising capability of continuous conditional diffusion denoising models, and propose a principled metric for measuring the gap between diffusion models and their idealised denoising path. Both the gap, and the metric that measures it have a strong mathematical motivation and are supported by empirical evidence. The authors show that this deviation is not caused by model capacity or amount of training data, and pose it as a difficulty of the model to bridge between various conditional areas of the space.

**Questions:**

1. How can SD be decreased for a specific task?
2. The paper explores SD only in the context of U-Nets, and MLP, how do you see that the architecture of the network impacts your metric? Do different architectures smooth differently?
3. Is the author's vision that because this gap (deviation) can’t be filled by increasing model capacity or the amount of training data, the problem cannot be solved?

**Ethical Concerns:**

["NO or VERY MINOR ethics concerns only"]

**Final Justification:**

I think this work has a lot of compelling insights that merit discussion at neurips 2025.

**Limitations:**

I will rate as borderline acceptance, but willing to increase in the case in which:
- The presentation of the paper would be improved (less-heavy first Figure 1, introduce important concepts earlier on).
- Related work is more rooted, and it explains why the classifier-free guidance paper is different [Bradley and Nakkiran, 2024] is different from the author's work.
- Perhaps a real-world example would be present, for example, in continuous control policies, and why it is important that the model doesn't “interpolate” between unseen conditions.

**Quality:**

2

**Strengths And Weaknesses:**

Please provide a thorough assessment of the strengths and weaknesses of the paper. A good mental framing for strengths and weaknesses is to think of reasons you might accept or reject the paper. Please touch on the following dimensions: Quality, Clarity,

-  Significance, and Originality. For more information, please see the NeurIPS 2025 Reviewer Guidelines (https://neurips.cc/Conferences/2025/ReviewerGuidelines). You can incorporate Markdown and LaTeX into your review. See https://openreview.net/faq.
Major Strengths:
-  The theoretical framework through which the importance of the introduced measure is very nice. The authors upper and lower bound total variation between the idealized and actual paths, with the introduced measure (Theorem 1).
- Self-guidance and linear interpolation of newly seen conditioned point is nicely proven in (Theorem 2-3)
The paper is theoretically sound and it makes a contribution in that regard.


Major Weaknesses:
- While the denoising mechanism fails only when the conditioning space is continuous and relevant applications for this are presented in the abstract (text-conditioned, or control policies), no experiments in this regard are present (only toy-examples).
- While the abstract is well-written, Figure 1 is too heavy to offer initial intuition on the problem. An unpopular task: t-SNE-condition MNIST generation is introduced without explanation, and then the similarity between Schedule Deviation (proposed method), and OT between samples produced by two methods (DDPM and DDIM). While the similarity is somewhat obvious it is not clear why the comparison is relevant (why compute OT between DDPM and DDIM samples?). The paper would benefit perhaps from a more intuitive first figure. Also, I am not sure why it is relevant to increase the size of the training data?
 - Related work discussed only in 6 lines  (from 93-99), is very sparse. It is mentioned that “[Bradley and Nakkiran, 2024] show that classifier-free guidance results in the diffusion not to be denoising anymore”, but it is not mentioned why their work is different.


Minor Weaknesses:
- It would be interesting to see how the samples from DDIM and DDPM actually differ (rather than the OT distance)
- I would introduce quicker why the disagreement between DDPM and DDIM is important and offer intuition of it.
- Same for Figure 2, it is not obvious why the SD is predictive of this divergence, it is not clear what is meant by trajectory N=4000 means (why is it relevant?). While I see value in the fact that the demonstration of the importance of SD is made through empirical effort, I believe that at least for Figure 1 the introduction is too heavy.

---

> ### Author Rebuttal · Authors · 2025-07-31
>
> We thank the reviewer for their feedback on the presentation of the paper and recommendations. Unfortunately we cannot update the paper until the camera ready deadline, but will detail the changes we intend to make given the reviewer’s constructive feedback.
>
> 1. **Experiments:**
>
>     Although we present only proxies for large-scale models/tasks, we are able to evaluate models up to the ~40 million parameter regime without issue. Thus we believe that with compute resources beyond those in academia this could be applied to smaller-sized text-conditioned models or robot foundation models (given perhaps several weeks compute time on a couple dozen GPUs).
>
>     In addition to the existing image experiments, we will replicate our methodology on either CelebA-Dialog or an annotated FFHQ which contains fine-grained labels for attributes such as age, degree of smile, amount of facial hair, etc. This will serve as a better proxy for text-based conditioning than t-SNE conditional MNIST without incurring the expense of evaluating models in the billion-plus parameter regime.
>
>     We originally ran experiments on the Push-T continuous control environment but found essentially no Schedule Deviation for the state-conditioned models, which we attribute to a lack of multi-modality in the action distributions. We believe this is due to overfitting of these large (50m+ parameter models) to very few (~200) human demonstrations, meaning these models behave more like an adaptive nearest-neighbor approximator than a multi-modal action model. Guidance between with identical, translated distributions simply shifts the mean and does not result in Schedule Deviation (see Appendix B for a discussion of Schedule Deviation for Gaussians), so it is unsurprising that we found no Schedule Deviation for essentially deterministic Diffusion Policies. We conjecture that for robot foundation models this “effective unimodality” is not present with large multi-task training datasets and image-conditioned models, although we acknowledge that additional studies of these models are needed.
>
>     When designing the maze navigation task, we explicitly introduced multi-modality in the action distributions by considering a weighted mixture of all possible paths to the target point when generating the data. Indeed, higher Schedule Deviation can be observed at “bifurcation” points in Figure 3 where the trajectory distribution is more multi-modal.
>
>     We would certainly like to run additional continuous-control experiments in other environments. However, we are unsure what an environment would look like where the diffusion policy is not effectively deterministic. We welcome suggestions from the reviewer on what environment they feel would be most appropriate.
>
> 2. **Presentation:**
>
>     We acknowledge that placing Figure 1 so early is too heavy and confusing for readers. We will move this to coincide with the main experiments to make the introduction more logical and make an easier-to-digest figure (perhaps based on the low-dimensional experiments in Section 4) to familiarize the reader to the problem setting we consider.
>
>    Through different values of “N,” the total amount of training data, we wanted to demonstrate that Schedule Deviation is present even when the model is given access to different quantities of training data at the “low-density” regions–i.e. the effect depends on the relative density rather than the total amount of training data and so we conjecture that Schedule Deviation cannot be ameliorated simply by scaling the amount of training data available. Note that we also provide ablations over the training data for Fashion-MNIST and the Maze Trajectories datasets in our experimental appendix.
>
>  3. **Prior Work:**
>
>     We will definitely expand on the prior work section. In particular, we will include significantly more details as to how our metric relates to work in unconditional models and how previously proposed metrics are either inappropriate for conditional models or otherwise differ from our proposed approach.
>
>     [Bradley 2024] observes that classifier free guidance does not yield a denoiser, although they have no methodology for quantifying the degree of this divergence. In our evaluation setup there is no classifier-free guidance–only a single conditional diffusion model. We use the observation of Bradley et al. that interpolating scores yields non-denoising probability paths to leverage our theoretical results in Section 4 to explain Schedule Deviation. We agree that this part of the prior work was not well-phrased and will better explain the relation to Bradley et. al. in our subsequent revision.
>
>  4. **Predicting OT Distance:**
>
>      Both DDPM/DDIM produce “reasonable” looking samples across the conditioning space. We conjecture that Schedule Deviation is predictive of the OT distance as DDIM/DDPM leverage the assumption that the flow is a denoiser differently in their sampling procedures. This is immaterial when the neural network well-approximates a denoiser (low Schedule Deviation) but can lead to divergence with high Schedule Deviation, which is precisely what we observe.
>
>  5. **Reducing SD:** We take no stance as to whether SD is “good” or “bad.” However, the widespread use of classifier-free guidance to improve generation quality suggests that any Schedule Deviation arising from “self-guidance” may in fact be an excellent inductive bias for models, in which case reducing Schedule Deviation may be detrimental. As we note in Remark 3.1, Schedule Deviation is not directly related to generation fidelity but rather measures the self-consistency of the reverse process.
>
>     Methods such as [Daras 2023] try to induce more denoiser-like behavior by augmenting with samples drawn from the reverse process. Augmenting with an entirely new, much larger, dataset using a pretrained model eliminates the interpolation ambiguity and specifies the correct interpolation. Whether this is beneficial likely depends on the inductive biases of both the model and the sampler used in this data re-generation process. This effectively would “fix” the interpolation to either the distribution generated by DDIM or DDPM.
>
>     Note that our metric of schedule deviation is different from the analogous quantities used in [Daras] or [Scarvelis] in that (1) we run a full reverse process and then re-noise our samples, whereas [Daras] uses only reverse process samples and (2) we more directly measure how the evolved density deviates from the ideal denoiser. It is possible to deviate from the ideal denoiser by adding “curl” components that do not change the probability path. By incorporating the divergence in our metric, we are agnostic to such changes.
>
>     We will expand on the above in the prior work section.
>
>
>  6. **Regarding Architectures:**
>
>     It would be quite surprising if different architectures interpolated the same way, as we observe empirical differences in the sample quality for different architectures. However both U-Nets and MLPs exhibited schedule deviation. Our paper conjectures this is because “reasonable” smooth interpolations between different denoisers are fundamentally non-denoising. Therefore any architecture which does not exhibit Schedule Deviation would likely need fundamentally very different and seemingly "unnatural" inductive biases compared to the architectures we consider here. We leave an in-depth exploration of the Schedule Deviation of different architectures to future work but believe it is of great interest, both theoretically and empirically.
>
> [Daras 2023] Daras, G., Dagan, Y., Dimakis, A., & Daskalakis, C. (2023). Consistent diffusion models: Mitigating sampling drift by learning to be consistent. Advances in Neural Information Processing Systems, 36, 42038-42063.
>
> [Scarvelis 2023] Scarvelis, C., Borde, H. S. D. O., & Solomon, J. (2023). Closed-form diffusion models. arXiv preprint arXiv:2310.12395.
>
> [Bradley 2024] Bradley, A., & Nakkiran, P. (2024). Classifier-free guidance is a predictor-corrector. arXiv preprint arXiv:2408.09000.

---

> > ### Comment · Reviewer_daH1 · 2025-08-04
> > **Thank you.**
> >
> > My comments have been addressed, and I have raised my score by a point.

---

### Note · Authors · 2025-08-15

Dear AC and Reviewers,

Thank you for taking the time to thoroughly review our manuscript and provide thoughtful and constructive feedback.
We look forward to incorporating the changes as discussed with the reviewers and hope we have clarified all the reviewers’ outstanding concerns.

---

### Decision · Program_Chairs · 2025-09-17

**Decision:**

Accept (poster)

**Comment:**

This paper presents a metric for quantifying how much the flow estimated by a diffusion model deviates from the flow defined by the minimizer of the DDPM objective [Ho et al. 2020] (aside - there's typo on line 128 where this is referred to as the DPPM objective). They show that trained diffusion models deviate from this idealized probability path, and offer an explanation for this phenomenon as a consequence of smoothness inductive biases. They also show that the score correlates with the differences between DDPM and DDIM samples. I agree with the reviewers that the paper is an insightful contribution that should be discussed at the conference. Beyond the clarifications brought up by the reviewers (e.g. emphasising the dependence on the estimated $p_0$, updates to Figure 1, etc.), I also agree with Reviewer zm5H that including a description of the algorithm in the main text would help many readers.